# The efficiency of secondary organic aerosol particles to act as ice nucleating particles at mixed-phase cloud conditions

**Wiebke Frey**[1,*], **Dawei Hu**[1], **James Dorsey**[1], **M. Rami Alfarra**[1,2], **Aki Pajunoja**[3], **Annele Virtanen**[3], **Paul Connolly**[1], and **Gordon McFiggans**[1]

[1]Centre for Atmospheric Science, School of Earth and Environmental Sciences, The University of Manchester, Manchester, UK
[2]National Centre for Atmospheric Science (NCAS), The University of Manchester, Manchester, UK
[3]Department of Applied Physics, University of Eastern Finland, Kuopio, Finland
[*]now at: Leibniz Institute for Tropospheric Research (TROPOS), Leipzig, Germany

*Correspondence to:* Wiebke Frey (frey@tropos.de)

**Abstract.** Secondary Organic Aerosol (SOA) particles have been found to be efficient ice nucleating particles under the cold conditions of (tropical) upper tropospheric cirrus clouds. Whether they also are efficient at initiating freezing at slightly warmer conditions as found in mixed-phase clouds remains undetermined. Here, we study the ice nucleating ability of photochemically produced SOA particles with the combination of the Manchester Aerosol and Ice Cloud Chambers. Three SOA systems were tested resembling biogenic/anthropogenic particles and particles of different phase state. These are namely $\alpha$-pinene, heptadecane, and 1,3,5-trimethylbenzene. After the aerosol particles were formed, they were transferred into the cloud chamber where subsequent quasi-adiabatic cloud activation experiments were performed. Additionally, the ice forming abilities of ammonium sulfate and kaolinite were investigated as a reference to test the experimental setup.
Clouds were formed in the temperature range of -20°C to -28.6°C. Only the reference experiment using dust particles showed evidence of ice nucleation. No ice particles were observed in any other experiment. Thus, we conclude that SOA particles produced under the conditions of the reported experiments are not efficient ice nucleating particles starting at liquid saturation under mixed-phase cloud conditions.

## 1   Introduction

Clouds and their feedbacks are major sources of uncertainty in future climate predictions. Aerosol particles, to a significant extent, determine the condensation of water to form liquid droplets and ice crystals. The transition into the ice phase is particularly important, e.g. for formation of precipitation, but is yet poorly understood in detail. While certain aerosol particles such as dust are known to be important ice nucleating particles (INP), others are highly abundant, yet their ice forming abilities remain poorly understood. One example for such particles are secondary organic aerosol (SOA; see also a recent review about the role of organic aerosol as INP by Knopf et al., 2018). They originate from biogenic and anthropogenic sources, e.g. from the oxidation of plant, biomass burning, and combustion emissions. SOA particles can exist in different phase states. The traditional understanding conceived them as homogeneous well-mixed liquids but they can occur in amorphous semi-solid or solid (also termed glassy) states (Virtanen et al., 2010). The state of the particles is dependent on the relative humidity and temperature (Koop et al., 2011; Berkemeier et al., 2014). The amorphous phase state of glassy particles has been shown to influence their ability to act as ice nucleating particles. For example Murray et al. (2010) have found glassy organic particles to be efficient INPs in the depositional mode in the tropical tropopause layer. In their experiments, Murray et al. (2010) found that glassy aerosol particles nucleated ice crystals at lower relative humidities (with respect to ice) than the same aerosol in a non-glassy phase state. Furthermore, fewer particles nucleated on the glassy particles, allowing higher in-cloud humidities. Several further studies have investigated the ice forming capability of different SOA particles (e.g. Prenni et al., 2009; Wang et al., 2012;

Ladino et al., 2014; Schill et al., 2014; Ignatius et al., 2016; Wagner et al., 2017) in the laboratory. The procedures for generation of the SOA particles as well as the methods to initiate ice nucleation do vary substantially among these experiments. E.g. SOA formation was initiated by dark ozonolysis (Prenni et al., 2009; Wagner et al., 2017) or photochemical reactions (Ladino et al., 2014; Ignatius et al., 2016), using gas-phase reactions (Wang et al., 2012; Ignatius et al., 2016) or aqueous processing (Wilson et al., 2012; Baustian et al., 2013; Schill et al., 2014). Ice nucleation was tested e.g. in expansion chambers (Murray et al., 2010; Wilson et al., 2012; Wagner et al., 2017), continuous flow diffusion chambers and flow tubes (Ladino et al., 2014; Ignatius et al., 2016), and in microscope systems (Wang et al., 2012; Baustian et al., 2013). Whether these formation pathways and ice nucleation initiation methods have an impact on the ice nucleation ability of the SOA particles is not clear. In addition to the different particle generation procedures, the resulting particle sizes vary, too, which certainly alters the ice nucleation capability (larger particles provide a larger surface area and are more likely to form ice). The findings of the above mentioned studies can be summarised as follows: Wang et al. (2012) and Ignatius et al. (2016) find that atmospheric SOA particles are potentially important for ice nucleation due to their semi-solid or solid phase states by investigating SOA from naphtalene and $\alpha$-pinene, respectively, whereas Ladino et al. (2014) and Wagner et al. (2017) found that $\alpha$-pinene SOA at first is an inefficient INP at cirrus temperatures, but after precooling of the SOA particles, ice nucleation ability is increased. Schill et al. (2014) found that semi-solid or glassy SOA from aqueous processing of methylglyoxal with methylamine is a poor depositional INP, however, Wilson et al. (2012) found other aqueous glassy aerosol to nucleate ice heterogeneously at temperatures relevant for cirrus formation in the tropical tropopause layer. All these studies investigated the ice forming abilities of the SOA particles at temperatures in the cirrus regime (i.e. below -40°C). Investigations of the ice nucleation potential of SOA in mixed-phase cloud conditions are scarce. One example of a study conducted at mixed-phase cloud temperatures is reported by Prenni et al. (2009). They looked at the ice nucleating ability of alkenes at -30°C and found them to be unlikely to participate in heterogeneous nucleation. However, they formed the SOA particles by dark ozonolysis of precursors. To our knowledge, the efficiency of photochemically produced SOA particles as ice nucleating particles under mixed-phase cloud conditions has not been determined. Modelling studies generally predict that SOA particles are efficient INP under cirrus conditions, i.e. low temperature and humidities (e.g. Koop et al., 2011; Berkemeier et al., 2014; Price et al., 2015) as the SOA particles are in a glassy phase state under those conditions. Furthermore, Shiraiwa et al. (2017) state that also in the middle troposphere SOA should be mostly in the glassy state, which may promote ice nucleation. They found SOA to undergo their glass transition above 2km altitude. Similarly, Mikhailov et al. (2009) state that a moisture-induced glass transition may play a role in the lower troposphere, depending on the relative humidity. Thus, SOA might play a role as INP in mixed-phase clouds, too.

The freezing and eventual sublimation of ice from the aerosol particle may change its properties (e.g. Adler et al., 2014): so called cloud processing. Thus, such a freeze-drying cycle might increase their ice nucleating abilities. Wagner et al. (2014) for example, found that pre-activated aerosol particles, i.e. temporarily cooled particles, have an increased ability for heterogeneous ice nucleation. Cloud processing also happens in warm clouds (e.g. Hoose et al., 2008), where the aerosol particle characteristics can be changed e.g through additional uptake of atmospheric gases and chemical reactions with the soluble part of the contained aerosol particle take place in the aqueous phase. Upon evaporation of the cloud, aerosol particles are re-emitted and these particles have changed chemical properties and are larger than the initial particles (Pruppacher and Klett, 1997); therefore, the aerosol size distribution is also affected by cloud processing. Comparison of the size distribution of interstitial aerosol within the cloud with the size distribution below the cloud clearly indicates that the processing of the aerosol through (nonprecipitating) stratus can lead to increased mass of the subset of particles which had served as cloud condensation nuclei (CCN; Hoppel et al., 1994).

This work aims to investigate the ice nucleating ability of photochemically produced SOA particles at mixed-phase cloud conditions in the Manchester Aerosol and Ice Cloud Chambers. Here, we report on the results of the measurements (Sect. 3, including the experimental setup), after an introduction of the chambers and instrumentation (Sect. 2). The results are discussed in relation to previous studies and their impact for atmospheric processes (Sect. 4) and summarised in the conclusions (Sect. 5).

## 2  Facilities and Instrumentation

For the current study the Manchester Ice Cloud Chamber (MICC) and Manchester Aerosol Chamber (MAC) were used, which are connected by a transfer pipe. The chambers and their instrumentation are described in the following. An overview is given in Fig. 1.

### 2.1  Manchester Aerosol Chamber (MAC)

The Manchester Aerosol Chamber (MAC) is a photochemical aerosol chamber comprising a $18m^3$ teflon bag (Hamilton et al., 2011; Alfarra et al., 2013) surrounded by a temperature and relative humidity controlled housing. The teflon bag is held by three frames such that the upper and lower frame can freely move to allow expansion and collapsing of the chamber during fill cycles or sampling (and thus

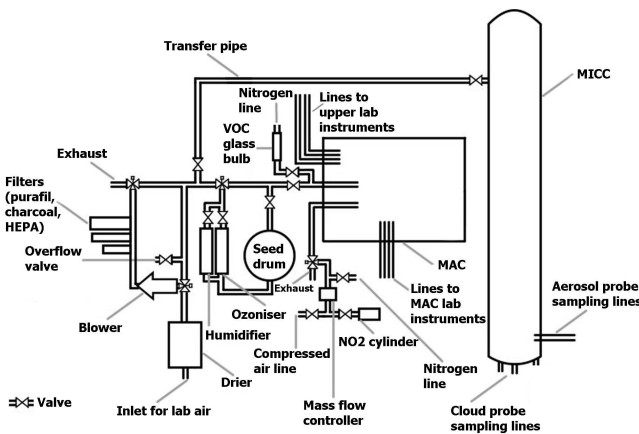

**Figure 1.** Drawing of the aerosol (MAC) and cloud (MICC) chambers, air system, connecting pipes, and various sampling lines.

removing air) from the chamber. Attached to the chamber is an air system that contains a series of filters (Purafil, Purafil Inc., USA; charcoal; HEPA, Donaldson Filtration, USA), a humidifier, an ozoniser, and a seed drum. $NO_x$ can be added
(as $NO_2$) as well as the target volatile organic compounds (VOCs) for SOA formation. The precursors are introduced through injection into a heated glass bulb and transferred into the chamber by a flow of filtered, high purity nitrogen (ECD grade, 99.997 %). Thus, the composition of gaseous
precursors and relative humidity can be controlled. Two 6kW Xenon arc lamps and further halogen bulbs are mounted on the inside of the bag's housing to simulate the solar spectrum and enable photochemistry. Furthermore, the housing is covered with reflective space blanket, in order to max-
imise the irradiance in the bag and to ensure even illumination. The Xenon arc lamps are mounted on two opposite sides of the enclosure at different heights. The illumination setup has been tuned to mimic the atmospheric actinic spectrum over the wavelength range 290–800nm. Air conditioning en-
sures that the chamber is kept at an operating temperature of typically 25°C during experiments (under illumination). Further details on the chamber and illumination can be found in Alfarra et al. (2013).

The aerosol chamber is equipped with a set of instruments to
25 measure temperature, humidity, aerosol number, particle size distribution, mass concentration, and chemical composition. Relative humidity and temperature are measured at the centre and on the side of the chamber by a dewpoint hygrometer, a thermocouple, and a resistance probe. A water condensation
particle counter (WCPC 3785, TSI Inc., USA; with a cut-off diameter at 5 nm nm) is used to measure the aerosol number concentrations. A Differential Mobility Particle Sizer (DMPS, custom made) observes particle size and mass distributions for sizes between 40–640nm with the sheeth flow
taken from the chamber as well. A chemiluminescence gas analyser (Model 42i, Thermo Scientific, USA) is used for

measuring NO and $NO_2$ mixing ratios and ozone was measured by a UV photometric gas detector (Model 49C, Thermo Scientific, USA).

A transfer pipe connects the aerosol chamber to the cloud  40
chamber (as indicated in Fig. 1). The pipe has a diameter of one inch and is approximately 33m long.

## 2.2 Manchester Ice Cloud Chamber (MICC)

The Manchester Ice Cloud Chamber (MICC) is a 10m high stainless steel tube of 1m diameter which is contained in  45
three cold rooms (spanning over three floors; Connolly et al., 2012; Emersic et al., 2015). The cold rooms can be temperature controlled from room temperature to approximately -50°C. Two scroll pumps are used to evacuate the chamber in order to form clouds in these experiments. The chamber can  50
be refilled, e.g. with filtered air or air from the aerosol chamber, via the transfer pipe (cf. Fig. 1).

MICC is instrumented to measure ambient conditions such as temperature, pressure, and humidity, and furthermore, cloud particle and aerosol particle concentrations. The following  55
instruments are in use: eight thermocouples (K type) at different heights, reaching alternating 10cm (thermocouples Tc1, Tc3, Tc5, Tc7) or 50cm (thermocouples Tc2, Tc4, Tc6, Tc8) into the chamber. The thermocouples were calibrated before the experiments at temperatures between -78.5°C (dry  60
ice) and 100°C (boiling point of water). The thermocouples have a time constant of about 60s, see appendix for more detail. A modified (for lower temperatures) Keller Lex1 pressure sensor with accuracy of 0.2%FS monitors the chamber pressure. Humidity can be measured at ambient pressure by  65
a CR-4 hygrometer. For the measurements of cloud particle properties such as number concentrations, size distributions, and shapes, a Forward Scattering Spectrometer Probe (FSSP Dye and Baumgardner, 1984) and a Cloud Particle Imager (CPI, version 1.0; Connolly et al., 2007) were deployed.  70

## 2.3 Cleaning procedures

In order to clean the chambers the attached air system accommodates three filters (see description in Sect. 2.1 and Fig. 1) to remove any particles and reactive gases.

The aerosol chamber is cleaned by several cycles of fill-  75
ing and flushing (at least 5 times) until the aerosol number concentrations stayed constant below $5cm^{-3}$ (dark concentration). The cleaning procedure includes flushing the pipes around the seed drum and ozoniser. After the last flushing cycle ozone is added into the chamber and left over night in  80
order to oxidise any left-over reactants. Typically for this purpose, ozone concentrations are about $500-600nmol\,mol^{-1}$. The content of the chamber is then replaced with clean air using a series of fill/flush cycles prior to the experiment. Once a week a SOA background experiment (see Sect. 3.2  85
for details) is conducted where the UV lights are switched on. These background experiments could be seen as 'harsh'

cleaning procedures, and the combination of the daily cleaning (fill/flush cycles and ozone overnight) and background cycles has been found to keep the chamber sufficiently clean. The cloud chamber is cleaned by repeated evacuations of the chamber to 200hPa followed by refilling from the air system with filtered air until aerosol concentrations stayed constant below $1 cm^{-3}$. The number of necessary cleaning cycles is dependent on the aerosol number concentrations left from a previous experiment.

As the cloud chamber is fitted with several outlets/openings, which for evacuation are sealed off, there are sources for leakages. These have been measured by evacuating the chamber to 200hPa and leaving it at this pressure for approximately two hours. A leakage rate of $0.12 hPa\,min^{-1}$ was found, which should only allow introduction of a small number of unspecified aerosol from the lab air. An estimate including previous leak checks and concurrent aerosol concentration measurements suggests an introduction of less than $1.5 cm^{-3}$ when leaving the chamber 5 minutes at 200hPa. (Less aerosol is introduced through leakages at higher pressures in the cloud chamber.) The leak check included the transfer pipe, i.e. the valve to the transfer pipe on the cloud chamber side was open and just closed at the entrance to the air system. The refilling of the cloud chamber with air from the aerosol chamber was performed as quickly as possible, to reduce time when MICC and transfer pipe are underpressured and can potentially be contaminated (typically started within 1 minute). A transfer, i.e. the refilling of the cloud chamber from the aerosol chamber to ambient pressure, takes about 10 minutes.

The air system itself is a complex system with various fittings; therefore, it presents a further potential source for contamination during transfers. This was tested by "clean bag" transfers, where both chambers are cleaned and a transfer is performed with (almost) particle free air from the aerosol chamber bag (measurements are shown in supplementary material).

In addition to the cleaning procedures, MICC is regularly defrosted to avoid build-up of ice on the sample line outlets to the cloud particle instrumentation that would eventually lead to particle losses.

## 3   Experiments

The experimental programme was constructed to test the efficiency of SOA to act as ice nucleating particles under conditions roughly resembling mixed-phase clouds where dust starts to become important as ice nucleating particle. Experiments were started at -20°C and close to water saturation to allow formation of liquid clouds which then glaciate. Thus, the setup also allows for testing of immersion freezing, i.e. whether the activated cloud droplets contain efficient (semi-) solid IN inclusions. Three different SOA systems were used to perform the experiments. They were cho-

sen to be representative of a typical range of SOA particles of varying sources found in the atmosphere, including anthropogenic/biogenic as well as particles of different phase state (here liquid and semi-solid particles; Virtanen et al., 2010, see also Sect. 3.1). The SOA were photochemically formed from the following precursors: $\alpha$-pinene (biogenic and semi-solid phase state), 1,3,5-trimethylbenzene (TMB; anthropogenic and semi-solid), and heptadecane (anthropogenic and liquid). In order to test the experimental setup, the experiments using SOA particles for cloud formation were accompanied with experiments using ammonium sulfate. Ammonium sulfate was chosen as it is a well known system that does not nucleate ice under the chosen experimental conditions. Furthermore, it can easily be tested with a model to assess whether the ammonium sulfate measurements are meaningful (see Sect. 3.4). A further control experiment was performed using dust (kaolinite) as this is known to be an ice nucleating particle at the given temperature. Some systems have been tested with two different pump speeds during the cloud activation experiments which alters the cooling rate. Cooling rates were approximately $10.1 K\,min^{-1}$ and $6.2 K\,min^{-1}$ for fast and slow pump speed, respectively. An overview of the performed experiments is given in Table 1.

### 3.1   Particle bounce measurements

Amorphous solid particles have been observed to bounce in an aerosol impactor, thus, the bounce of particles can be used to infer their phase state (Virtanen et al., 2010). In a previous experiment using the combination of MAC and MICC, transfer experiments on some of the same (and some additional) systems were performed and the phase state of particles were determined by particle bounce measurements (see Saukko et al., 2012b, a, for a detailed description of the methods). Here an upgraded version of the bounce system, the Aerosol Bounce Instrument (ABI Pajunoja et al., 2015), was employed. ABI consists of a particle size selection unit (neutralizer containing bipolar 210Po strip and Vienna type long DMA), a humidification unit (Permapure PD-240-12SS, Nafion multitube), impactor unit (MOUDI stage #14 with upstream pressure, $p_{initial} = 0.85 bar$, and downstream pressure, $p_{final} = 0.7 bar$, leading to a cut-off aerodynamic diameter $d_a = 67.09 nm$, and two CPCs (TSI, model 3010) for measuring the particle number concentration before and after the impactor. ABI determines a bounced fraction (BF) of particles which is used as an indicator of the phase state of particles; particles with BF 0 are mechanically liquid whereas particles with $0.1 < BF < 1$ are mechanically solid or semi-solid. Calculation and calibration of bounce measurements are described in more detail by Saukko et al. (2012a).

In Fig. 2, the bounced fractions measured for TMB, heptadecane, $\alpha$-pinene/limonene and limonene experiments as a function of RH are shown. In general higher bounce fractions are found at lower relative humidity. For TMB, limonene and $\alpha$-pinene/limone cases the bounced fractions are high at dry

**Table 1.** Overview about the conducted experiments, indicating pump speeds, number of cloud runs performed, times of cloud droplet activation in seconds from start of cloud expansion, and initial $S_{ice}$ and initial $S_{water}$ for first/second(/third) cloud run.

|        | system              | pump speed     | # runs | onset of main activation | initial $S_{ice}$ [%] | initial $S_{water}$ [%] |
|--------|---------------------|----------------|--------|--------------------------|-----------------------|--------------------------|
| Exp 1  | clean               | fast           | 3      | 23s/19s/19s              | 106/120/112           | 82/98/92                 |
| Exp 2  | ammonium sulfate    | fast           | 3      | 21s/16s/18s              | 118/125/118           | 97/102/96                |
| Exp 3  | SOA background      | fast           | 2      | 19s/20s                  | 118/125               | 96/102                   |
| Exp 4  | $\alpha$-pinene     | fast           | 2      | 19s/24s                  | 127/126               | 104/103                  |
| Exp 5  | SOA background      | fast           | 2      | 19s/20s                  | 117/116               | 95/95                    |
| Exp 6  | heptadecane         | slow           | 2      | 31s/29s                  | 118/118               | 96/97                    |
| Exp 7  | TMB                 | slow           | 2      | 34s/32s                  | 125/114               | 102/93                   |
| Exp 8  | $\alpha$-pinene     | slow           | 2      | 35s/30s                  | 124/123               | 101/100                  |
| Exp 9  | ammonium sulfate    | slow           | 2      | 29s/27s                  | 122/117               | 100/95                   |
| Exp 10 | heptadecane         | fast           | 2      | 21s/21s                  | 117/117               | 96/96                    |
| Exp 11 | dust (kaolinite)    | fast and slow  | 2      | 21s/39s                  | 126/119               | 103/98                   |

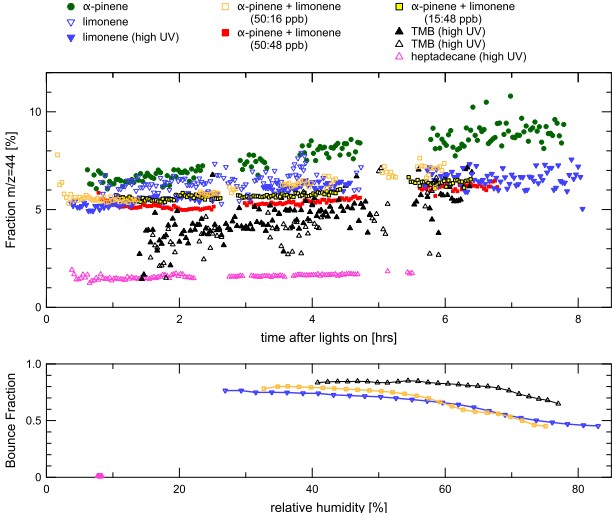

**Figure 2.** Composition and bounced fraction of SOA particles in MAC. Open symbols represent data collected in experiments where there was no quantitative ice nucleation data. Panel a shows the mass fraction of the AMS fragments measured at m/z=44 throughout their growth in the aerosol chamber. Panel b shows bounced fractions for heptadecane, $\alpha$-pinene/limonene and limonene experiments as a function of RH at the latest available time before transfer. In general higher bounce fractions are found at lower relative humidity. It can clearly be seen that SOA from the heptadecane experiment exhibit very different behaviour, maintaining a bounced fraction of less than 0.012 even at the lowest available RH. All other bounced fraction data is similar to the other examples shown. The heptadecane experiment also shows a much lower m/z=44 fraction than in other experiments.

conditions and stay elevated (BF > 0.4) up to RH 80% indicating semi-solid phase of the particles at elevated humidities. It can clearly be seen that SOA from the heptadecane experiment exhibit very different behaviour, maintaining a bounced fraction of less than 0.012 even at the lowest available RH indicating liquid phase state already at dry conditions. Similar liquid behaviour was observed for heptadecane earlier by Saukko et al. (2012b). The heptadecane experiment also shows a much lower m/z=44 fraction than in other experiments. Even though these bounce measurements were not performed simultaneously to the ice nucleation experiments we report here, the bounce measurements were conducted under very similar conditions in the same chamber using the same particle generation procedures. Therefore, we believe that the results are applicable to the SOA particles formed in our recent experiments. Unfortunately, no bounce measurements with pure $\alpha$-pinene were performed, however, several studies (e.g. Saukko et al., 2012a; Pajunoja et al., 2015; Ignatius et al., 2016) have shown that $\alpha$-pinene SOA is semi-solid under the conditions relevant here.

### 3.2 Experimental design

An experiment always followed the outlined procedure: After careful cleaning of the chambers and the air inlet system, the desired particles are created in MAC. To prepare the system for injection of relevant gases for particle formation in MAC the volatile organic compound (VOC) injection glass bulb is heated and continuously flushed with nitrogen. The precursors for the SOA are injected into the glass bulb in form of high purity liquids, where they evaporate immediately. The vapourised VOCs and $NO_x$ are then injected during the last filling of the MAC air bag. By filling through the humidifier water vapour is added. Mixing within the bag is ensured by the main filling air stream. Photochemistry is started by switching on the lights. Ozone is injected as well just after the lights are switched on as a source of OH to speed up aerosol nucleation and to increase particle numbers. After sufficient time for the photochemistry, the lights are switched off and the cloud chamber is evacuated to prepare for the transfer. Table 2 shows the initial concentrations and other chamber conditions used for the formation of SOA particles. A typical development of the formation of a SOA system is shown in Figure 3.

When total aerosol particle mass reached equilibrium in

**Table 2.** Initial nominal conditions for SOA formation in the aerosol chamber. For a description how the VOCs are injected into the chamber see Sect. 3.2

| Experiment | precursor | nominal VOC | initial conditions after injection | | | | |
| | | mass | $NO_x$ | $O_3$ | VOC/$NO_x$ | $Conc_{CPC}$ | $Mass_{DMPS}$ |
| | | [nmol mol$^{-1}$] | [nmol mol$^{-1}$] | [nmol mol$^{-1}$] | | [cm$^{-3}$] | [µg m$^{-3}$] |
| Exp 1 | clean | - | - | - | - | <1 | - |
| Exp 2 | ammonium sulfate | for 60sec | 9.8 | 1.2 | - | 8477/4288 | 0.82 |
| Exp 3 | SOA background | - | 50.3 | 5.8 | - | 1.9 | 7.8e-5 |
| Exp 4 | $\alpha$-pinene | 250 | 38.9 | 10.3 | 6.4 | 25173 | 16.2 |
| Exp 5 | SOA background | - | 43.2 | 13.8 | - | 3087 | 0.03 |
| Exp 6 | heptadecane | 500 | 32.9 | 32.1 | 15.2 | 4289 | 4.2 |
| Exp 7 | TMB | 500 | 55.0 | 21.7 | 9.1 | 4595 | 0.51 |
| Exp 8 | $\alpha$-pinene | 100 | 24.0 | 1.7 | 4.2 | 9990 | 4.3 |
| Exp 9 | ammonium sulfate | for 30sec | 5.4 | 0.1 | - | 4824 | 0.72 |
| Exp 10 | heptadecane | 500 | 38.8 | 16.2 | 12.9 | 6035 | 2.2 |
| Exp 11 | dust (kaolinite) | not filled via MAC | | | | | |

**Table 3.** Conditions in the aerosol chamber shortly before transfer to the cloud chamber and aerosol properties after transfer in MICC.

| Experiment | precursor | time lights on | conditions: at transfer in MAC | | | | | after transfer in MICC | |
| | | | NO | $NO_2$ | $O_3$ | $Conc_{CPC}$ | $Mass_{DMPS}$ | $Conc_{SMPS}$ | $Mass_{SMPS}$ |
| | | [h] | [nmol mol$^{-1}$] | [nmol mol$^{-1}$] | [nmol mol$^{-1}$] | [cm$^{-3}$] | [µg m$^{-3}$] | [cm$^{-3}$] | [µg m$^{-3}$] |
| Exp 1 | clean | - | - | - | - | 0.3 | - | 10.8[!] | - |
| Exp 2 | ammonium sulfate | - | 9.2 | 0.3 | 0.7 | 3343 | 0.6 | 1993[!] | - |
| Exp 3 | SOA background | 3:25 | 11.3 | 30.6 | 2.2 | 1.4 | 0.1 | 27.2[!] | (<) |
| Exp 4 | $\alpha$-pinene | 5:12 | < | 19.2 | 9.3 | 9255 | 83.9 | 8483 | 92.1 |
| Exp 5 | SOA background | 5:36 | 1.0 | 25.5 | 17.8 | 1190 | 0.4 | 1082 | 0.6 |
| Exp 6 | heptadecane | 5:34 | < | 8.4 | 54.9 | 2143 | 112.6 | 2021 | 95.4 |
| Exp 7 | TMB | 5:47 | 3.1 | 34.1 | 36.4 | 1730 | 3.8 | 1737 | 8.4 |
| Exp 8 | $\alpha$-pinene | 5:30 | < | 9.7 | 9.8 | 2787 | 10.4 | 2817 | 14.9 |
| Exp 9 | ammonium sulfate | - | 9.2 | 1.7 | 1.5 | 2077 | 0.3 | 1996 | 0.5 |
| Exp 10 | heptadecane | 5:04 | < | 15.6 | 43.5 | 3353 | 92.9 | 2495 | 70.1 |
| Exp 11 | dust (kaolinite)* | - | 8.2 | 6.0 | 1.1 | 1.3 | <* | 554 | 4.57 |

[!] SMPS data not available (too low number concentrations) or faulty, concentrations taken from CPC
* dust injected into MICC directly
< below detection limit

MAC, a transfer was performed from MAC to MICC. For the transfer MICC was evacuated to 200hPa and then refilled from MAC to ambient pressure. Thus, the desired aerosol population is transferred into the cloud chamber and slightly diluted by the remaining air in MICC, i.e. approximately 8m$^3$ air from MAC is transferred to MICC and mixed with the approximately 2m$^3$ remaining clean air. Table 3 specifies the conditions in MAC short before and in MICC after the transfer. The dilution leads to a tendency for the semi-volatile components of the aerosol to evaporate. Since the air is cooled at the same time, there also is the opposing tendency of condensation and any semi-volatile component in the aerosol will have a tendency to transfer between phases accordingly. As there is no humidity nor organic vapour measurement during the transfer, the exact state of the aerosol is unknown. The transfer was then followed by measuring aerosol total number concentration and size distribution in MICC (using the CPC and SMPS). Temperatures in MICC fluctuate during a transfer, decreasing during chamber evacuation and increasing during refill, even above the target temperature of 253K, as the aerosol chamber is operated at room temperature, and the transferred air needs time to cool. Therefore, further aerosol measurements were obtained after MICC temperatures settled back to the target temperature. Comparison of aerosol size distributions from just after transfer and just before a cloud activation experiment, i.e. after the temperatures have settled, reveal that there is no significant change. Therefore, we conclude that either no significant evaporation and condensation took place or both effects cancel each other. As the chamber walls were ice coated, any humidity in excess of RH$_{ice}$ would have condensed onto the chamber walls following the transfer. However, given the dimensions of the chamber and the low speed of the diffusion process supersaturation was not depleted until the start of the experiments: Immediately following the later aerosol measurements, a cloud activation run was performed, i.e. MICC was pumped down to 700hPa with the cloud probes sampling from the chamber as well. Two different pump speeds were tested, a faster pump speed using both main pumps and the pumps attached to the cloud probes, and a slower pump speed using one of the main pumps only in addition to the cloud probes. A faster pump speed, assuming adiabaticity,

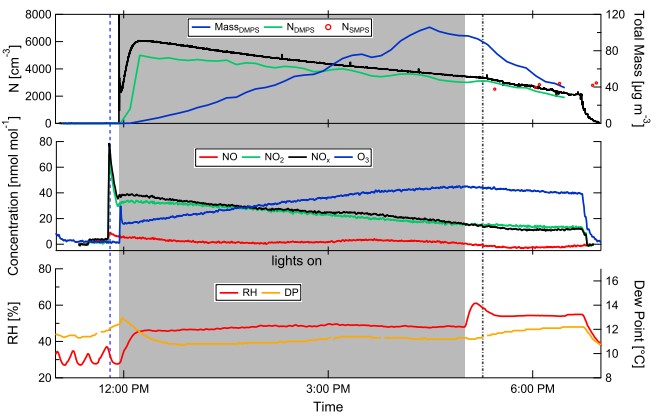

**Figure 3.** Development of SOA particles in the aerosol chamber. The blue dashed line indicates the injection of the SOA precursors, the grey dash-dotted line the beginning of the transfer to the cloud chamber. The grey shading indicates the time with the lights switched on. This example shows the development of SOA from the heptadecane precursor (experiment 8, see Table 1).

will lead to a faster cooling rate and higher supersaturations. MICC was refilled again from MAC, to avoid further dilution of the aerosol by mixing with filtered air. Again, aerosol number concentrations and size distributions were measured in MICC two times and a further cloud expansion was performed. If there was still enough air remaining in MAC, it was used to refill MICC which allowed a third cloud run. Additionally, cloud activation experiments on background transfers were performed. The background experiments normally contain all of the chemical substances as in a typical SOA experiment, with the exception of the main precursor. They are conducted to quantify the contribution of background VOCs and oxidants to the overall SOA formation and to ensure that the SOA formed during actual experiments is a result of the oxidation of the precursors being studied (i.e. not originating from compounds coming off the chamber walls or from the air used to fill the chamber).

## 3.3 Results

Instead of showing data for all cloud expansions here, we will only illustrate in detail two examples, one using heptadecane as precursor and one $\alpha$-pinene, as the latter looks fairly similar to the TMB experiment. For the sake of completeness, figures for all other cloud activation runs can be found in the supplementary material. An example for a cloud activation run on SOA formed on heptadecane precursor with the faster pump speed is shown in Fig. 4. The uppermost panel shows the aerosol size distribution measured by the SMPS prior to the chamber evacuation, along with the numbers for total aerosol concentration as observed by the SMPS. Agreement of the size distributions observed after transfer and before expansion shows that there is no significant alteration of the

aerosol size distribution as a result of the time spent in MICC while the temperatures settle. The mean mode diameter of the aerosol is located at about 370nm, while the second mode diameter is at about 200nm. Thus, these aerosol particles are large enough to potentially act as ice nucleating particles. Simultaneously with the aerosol measurements, humidity was scanned in MICC. These observations show that MICC was almost saturated with respect to water (RHw = 96%, supersaturated with respect to ice), the dew point was at 252.6K after the transfer and at 252.7K before the expansion. The further panels in Fig. 4 show the time series of the cloud development, with the size distribution and mean volume diameter (MVD) of cloud particles (panel b) observed by the FSSP, total water content (TWC) and number concentration (N; panel c), pressure and temperature (panel d), and some example images taken by the CPI (panel e). Cloud particles observed just at the start of the expansion potentially stem from opening the valve to the instrument inlet and should not be considered.

At the beginning of the cloud activation run (first 20 seconds) a small number of aerosol particles (approx. $30\mathrm{cm}^{-3}$) activate to cloud particles with sizes mostly below 10μm. The main activation takes place at 21 seconds, apparent from the cloud particle size distribution time series. High numbers of small hydrometeors are observed that subsequently grow to slightly larger sizes (cf. yellow colours showing the main particle size mode and superimposed mean volume diameter (MVD, light blue) in the plot). The CPI only detected spherical particles during the expansion. We would expect that potential ice particles would grow to larger sizes than the observed sizes. Taking the sizes and the imaged spherical shapes into consideration, it can be reasonably assumed that the observed particles were water droplets.

Fig. 5 shows the example of a cloud activation run performed with a slow pump speed on SOA generated from $\alpha$-pinene. The mean mode diameter of the aerosol is at approximately 210nm. Activation of aerosol to cloud particles starts at 30s into the cloud run. Particle sizes stay below 20μm, the mean volume diameter reaches about 9μm at about 90 seconds into the cloud activation run and stays fairly constant until the cloud diminishes. Only one spherical particle was imaged by the CPI. Given that there was no further growth in particle size and particle sizes are rather small, we conclude that the particles were in the liquid phase and not frozen. In the case of frozen particles, we would have expected quicker growth to larger sizes, e.g. as in the dust example below.

In order to show that ice can be formed under the experimental conditions, kaolinite dust particles were injected into the cloud chamber. The kaolinite dust (KGa-1b) was injected into the cloud chamber with the help of a dust generator (PALAS RBG1000) directly attached to the chamber (not via the air system). To ensure proper mixing of the dust and air in the chamber an evacuation to 700hPa was performed directly after the injection. The results of the dust run using the high pump speed is shown in Fig. 6. Dust particles of a wide

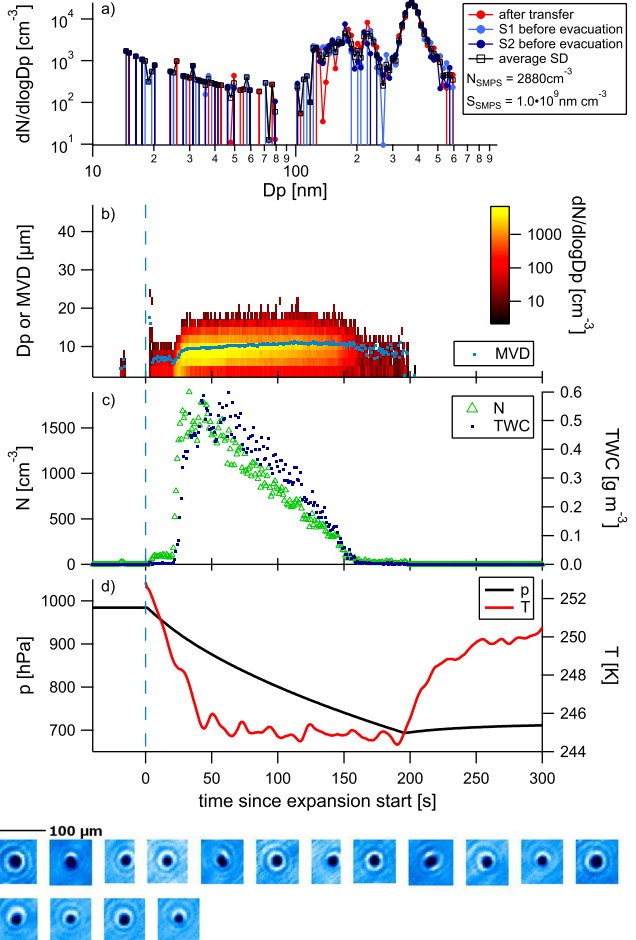

**Figure 4.** Example of a cloud evacuation performed on heptadecane precursor aerosol (experiment 10, see Table 1). The uppermost panel shows the SMPS size distributions obtained before the expansion, followed by timeseries of FSSP measurements, size distribution and mean volume diameter (MVD, panel b), total water content (TWC) and number concentration (N, panel c), and temperature and pressure (panel d) during evacuation. Below the time series images captured by the CPI are shown. This was the second run on the aerosol population with a fast pump speed.

range of sizes were present (see SMPS size distribution). At first, small particles with mean volume diameters between 4µm and 10µm were observed that were presumably large (swollen) dust particles. Upon activation at 21 seconds, small particles activated in the droplet mode (cf. yellow colours in the size distribution time series), followed by particle growth and a diminishing of the small droplet mode. The drop of cloud particle numbers at about 80s into the cloud activation run is caused by the growths of the larger ice particles, at the expense of the small droplets (Wegener-Bergeron-Findeisen process). The CPI images show the presence of non-spherical particles, i.e. ice particles.

The two successive cloud activation runs in each exper-

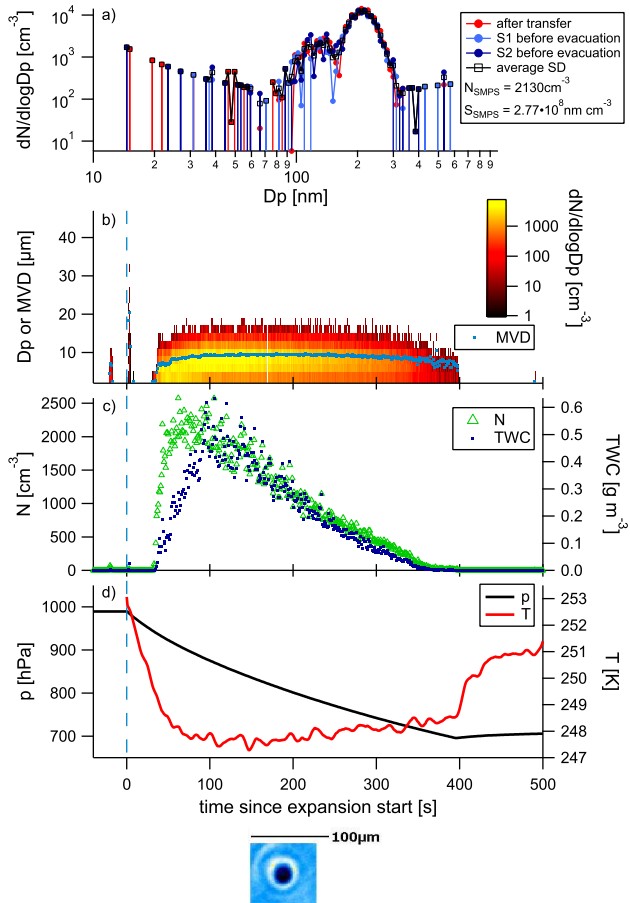

**Figure 5.** As Fig. 4 but using $\alpha$-pinene as SOA precursor (experiment 8, see Table 1). This evacuation was the second run on the aerosol population, performed with a slow pump speed. Only one image was sampled by the CPI.

iment can be used to look at activated fractions in order to see whether the aerosol properties change after one activation/deactivation cycle (cloud processing). The activated fraction here is simply calculated by dividing the pre-expansion aerosol number concentration by the peak cloud particle number concentration. Furthermore, when a second experiment is available with a different pump speed, this can be used to determine the effect of the cooling rate on the activation of the aerosol. Fig. 7 shows the activated fractions of five experiments: the already shown heptadecane (upper left panel), $\alpha$-pinene (middle right panel), and dust (lower right panel) activation runs accompanied by the respective other runs in the same experiments, plus a further heptadecane experiment with altered pump speed (lower left panel) and the TMB experiment (upper right panel). Instrument error margins may lead to an activated fraction of more than 1. The fast pump speed heptadecane experiment shows no significant cloud processing, activated fractions of both runs are very similar. In the slower pump speed experiment, however,

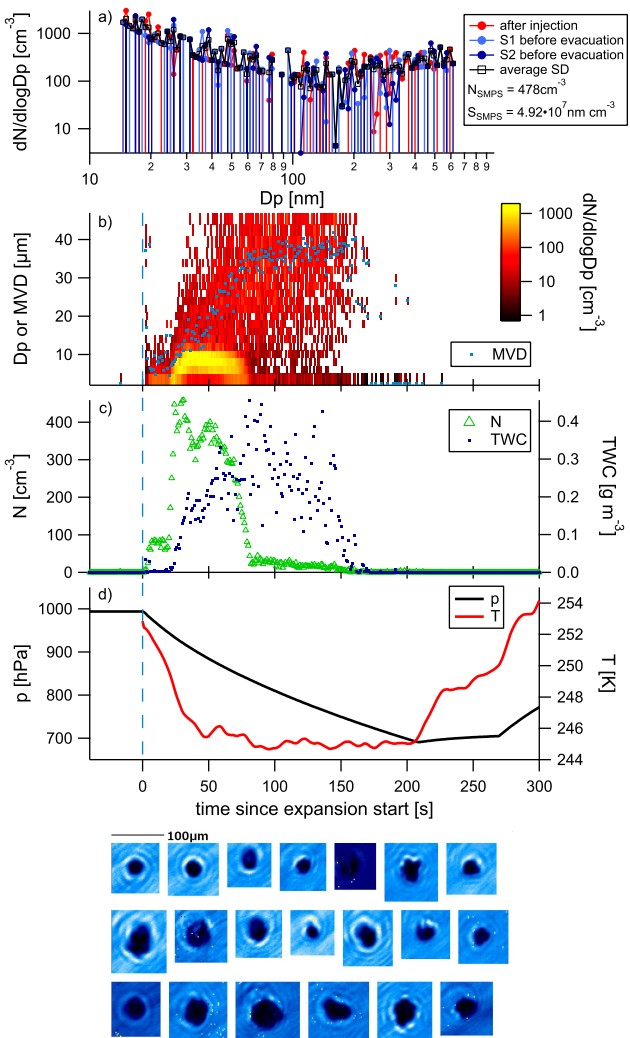

**Figure 6.** Control experiment using kaolinite dust (experiment 11, see Table 1). Panels as in Fig. 4, the data shown here stem from the first run, using a high pump speed. The data show formation of ice in a second mode, and the decrease and almost disappearance of particle numbers of smaller drops over time (Wegener-Bergeron-Findeisen process).

the second heptadecane run shows a higher activated fraction than the first run, though initial ice supersaturation are the same and temperatures in both runs are within $0.2°C$. Thus, the aerosol becomes more efficient at activating to cloud droplets. The first run here exhibits lower activated fractions as the fast pump speed runs, the second run peak activated fraction is about the same as in the fast pump speed runs. The $\alpha$-pinene slow pump speed experiment shows the opposite behaviour, the second cloud run has slightly lower activated fractions as the first. The same is true for the TMB slow pump speed runs. However, the initial temperatures differ by about $0.7°C$ in the TMB case resulting in a less strong temperature drop during the expansion, and also the initial

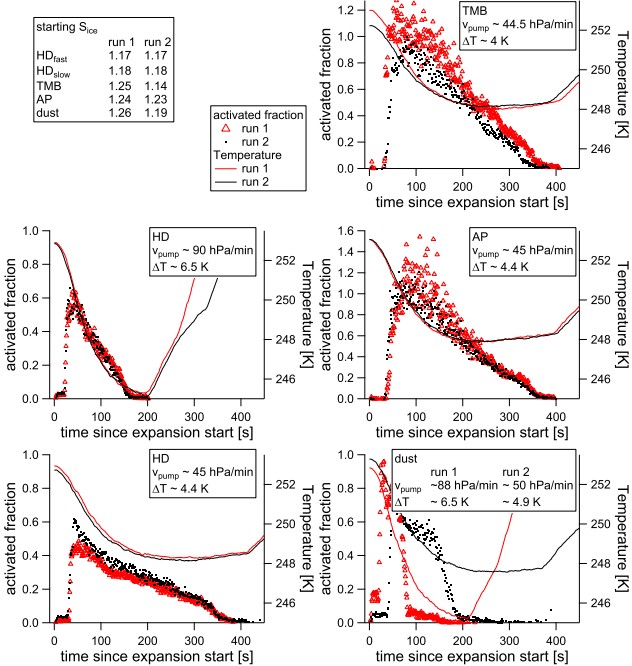

**Figure 7.** Activated fractions of aerosol to cloud particles for heptadecane (HD; left hand side), TMB (TMB; upper right), $\alpha$-pinene (AP; middle right), and dust (lower right). The first cloud evacuation run is shown by the red triangles, the second run by black dots. Pump speed and temperature drop during evacuation are given in the legends.

ice supersaturation differs. Caution needs to be taken with the outliers in the first $\alpha$-pinene run, which might be due to oversampling in the FSSP. The dust runs were performed at different pump speeds, the second run using the slower pump speed shows slightly lower activated fractions than the fast pump speed run.

## 3.4 Model comparison for ammonium sulfate control experiment

The Aerosol-Cloud and Precipitation Interactions Model (ACPIM; Connolly et al., 2012) has been chosen for testing of the experimental data of the ammonium sulfate control runs. The model is adapted to be used with chamber measurement data as the ones reported here. The observed temperature and pressure curves as well as the initial relative humidity and aerosol size distribution and number concentrations are used to initialise the model. Figure 8 shows the results of the ACPIM simulation in comparison to the measurements of cloud droplet number concentrations, LWC, and size distribution of the cloud activation run 2 of Experiment 9. The model predicts complete activation of the aerosol particles. The measurements show some outliers which might indicate that the capacity of the instrument for measuring particle numbers has been reached, i.e. too many particles lead

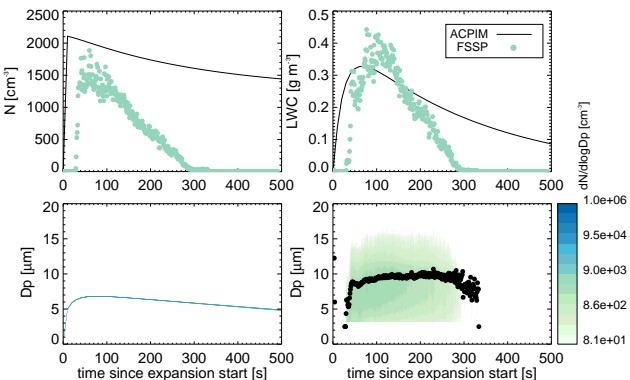

**Figure 8.** Comparison of ammonium sulfate measurements (Exp. 9, run2, see Table 1 and ACPIM simulation of number concentrations (upper left), LWC (upper right), size distribution and mean volume diameter (lower panels, measured right, simulated left).

to overcounting which is apparent in the outliers. Apart from the outliers, the measured cloud particle numbers are a little smaller than the modelled numbers, again in high concentrations the instrument is prone to measurement coincidence errors, not only overcounting but also multiple particles in the sample volume at one time leading to a general undercounting. As all aerosol particles in ACPIM activate, they grow subsequently into a very narrow size distribution as theory predicts. The simulated sizes are somewhat smaller than the mean volume diameter measured by the FSSP. This could be the reason for the smaller LWC predicted than measured as well. The simulation predicts cloud particle appearance earlier than the observations show, this is thought to be due to boundary layer effects in the cloud chamber. Due to e.g. wall heating, the air next to the walls may stay cloud free for longer than the interior of the chamber where the cloud forms earlier on. As the sample lines are attached to the bottom and will first suck air from the boundary layer, it takes time for the cloud to extend downwards to the bottom and be sampled, see also discussion in Möhler et al. (2003) for similar effects in the AIDA cloud chamber.

The observed cloud particle size distribution is wider than the simulated one. This could be due to effects of non-uniformity in temperature and humidity that lead to broadening of the size distribution. For example, as the chamber walls will stay at close to their initial temperature while the chamber centre cools adiabatically during an expansion, wall heating effects can create a temperature gradient within the chamber (warmer towards the walls and colder towards the chamber centre), which will induce a humidity gradient as well. The incoming air stream causes mixing which on the one hand side can reduce this gradient, but on the other side can induce inhomogeneous temperature and humidity fields leading to variations in the activation and growth of the cloud particles. However, the simulation and measurements are similar

enough to conclude that the measurements behave reasonably as expected.

## 4   Discussion

The goal of this study was to characterise the ice nucleating abilities of photochemically produced SOA under mixed-phase cloud conditions. Three different precursors were chosen for SOA formation in this study, to cover biogenic and anthropogenic as well as particles in different phase states (here semi-solid and liquid). The photo-oxidised SOA particles were transferred into a cloud chamber and their ice nucleating abilities were tested at temperatures of -20°C to about -28°C. The chamber was close to water saturation at the start of the measurements, thus, providing the environment for cloud formation. The most important finding from this study is that the SOA particles tested were not efficient ice nucleating particles at the chosen temperature and relative humidity range (i.e. in the mixed-phase clouds regime). While the sensitivity runs performed on kaolinite dust particles, clearly show nucleation of ice, ice nucleation was not measurable in any of the other SOA or ammonium sulfate runs. Generally, measurements of ice nucleation (or lack of ice nucleation) at temperatures above the cirrus regime are scarce. One example is the study by Prenni et al. (2009) who also found no measurable ice nucleation in continuous flow chamber measurements at -30°C. However, the residence time of the aerosol particles in Prenni et al.'s measurements in the continuous flow chamber are in the order of seconds, compared to several minutes here. Furthermore, they formed the SOA particles through dark ozonolysis of precursors using excessive amounts of ozone, whereas in this study photo-oxidation of precursors with less ozone were used. While our results are in line with the findings e.g. from Prenni et al. (2009) and Schill et al. (2014) who concluded that their aqueous SOA is a poor depositional ice nucleus though the aerosol particles were in a semi-solid or glassy phase state, other studies at lower temperatures found SOA to be an efficient INP (e.g. Wang et al., 2012; Ignatius et al., 2016) and thus potentially important for atmospheric ice nucleation. Further studies found that SOA from $\alpha$-pinene is initially an inefficient INP at cirrus cloud conditions and shows an increased ice nucleation ability when precooled or preactivated, i.e. after cloud processing where the aerosol first activated to supercooled cloud droplets and then froze homogeneously (Ladino et al., 2014; Wagner et al., 2017). These and other studies on the ice nucleating ability of SOA particles are summarised in Hoose and Möhler (2012) and Knopf et al. (2018). It has to be stressed that in the above mentioned studies and our study the SOA generation procedures and conditions can vary significantly, as well as the methods used to measure ice nucleation, see the Introduction for details. The precise impact the different SOA formation procedures and conditions have on the

ability of the SOA as INP, e.g. whether SOA formed by dark or light ozonolysis would behave differently if temperatures and humidities during the generation are the same is not clear. In order to help initiate the photochemistry $NO_x$ was used in the SOA formation here, which might be different to other SOA formation conditions. However, to the authors knowledge, there is no systematic study of the effect of $NO_x$ on particle composition or properties, so whether this has an effect on the ice nucleating ability of the formed SOA is not known. It should be noted though, that also the O:C of produced SOA was relatively low in our experiments. As shown in Pajunoja et al. (2015) the particle viscosity (or bounce behaviour) at elevated RHs depends strongly on O:C. This is because the particle hygroscopicity (i.e. the particle phase water content at certain humidity) increases with O:C. Hence, not only the $NO_x$ might affect the composition or viscosity, but also the oxidation conditions more generally. Furthermore, the main sizes of the produced SOA particles show large differences, ranging from below 100 nm to 40 μm, which will impact on their ice nucleating ability due to the available surface area. While most studies generate their SOA particles at room temperature and (almost) 0% RH, only Wang et al. (2012) and our experiments stand out with 35±5% and 45-50% RH, respectively. SOA generated at different temperatures and humidities might exhibit different IN potential, as such conditions impact on the phase state of the particles (see e.g. discussion in Berkemeier et al., 2014). We aimed at generating SOA in our experiments under conditions which are close to realistic conditions in the lower boundary layer, where precursor gases for SOA formation are emitted (here: room temperature and relative humidities wrt water of 45-50%). Berkemeier et al. (2014) further state that humidification in typical atmospheric updrafts (or cloud chamber experiments) may be fast enough to cause a difference in phase state from that of equilibrium, as the time for diffusion of the water into the particle is longer than the time for humidification. This can result in a particle that has a liquid outer shell but still contains solid inclusions/a solid core even at RH above the quasi-equilibrium glass transition. Thus, SOA partciles could potentially act as immersion freezing nuclei in conditions where they are supposed to be liquid, if in equilibrium. Even if Mikhailov et al. (2009) and Shiraiwa et al. (2017) are correct in assuming that glass transition plays a role at ambient temperatures in the lower troposphere (above roughly 2 km), SOA particles according to this study and Prenni et al. (2009) will not be (efficient) ice nucleating particles, neither as depositional nor as immersion freezing nucleus. Thus, even though they are abundant, SOA particles might not play a role in ice formation at lower altitudes.

There seems to be a twofold activation of cloud droplets in the heptadecane experiments. Upon the onset of activation a few seconds after start of the evacuation, a smaller amount of cloud particles are observed with the FSSP with a low LWC, until the second mode of activation commences and larger numbers of particles activate and the LWC peaks. The mean volume diameter of the cloud particles in the first activation mode is a little smaller than in the second mode, though not always clearly apparent. Also in the kaolinite dust experiments two modes were observed in the size distribution. Here, however, it is likely that the first mode comprises larger (possibly swollen) dust particles and not cloud particles. The latter appear in the second mode. The experiments with SOA particles from $\alpha$-pinene and TMB precursors show a different behaviour with only the main activation mode. The aerosol particles formed from these precursors are both in a semi-solid phase state, i.e. more viscous than the particles generated from heptadecane at the same temperature and relative humidity. Thus, it takes a longer time for these viscous SOA particles to take on water vapour and grow due to diffusion limitations. A speculative explanation for the behaviour of the heptadecane SOA particles could be that the particles, or rather a subset thereof, already start activating during the transfer. The relatively warm air from the aerosol chamber flows into the cold cloud chamber and starts cooling, the relative humidity increases accordingly. Some vapour will condense onto the walls, but the relative humidity might increase enough to start activation of aerosol particles that already made it into the chamber. These activated particles then grow, while further later arriving particles may stay unactivated. When the evacuation of the cloud chamber starts, cooling will be much higher and all other heptadecane particles will activate as well. Indeed, an effect of different temperatures in the chambers or sampling lines had been found and discussed by e.g. Ignatius et al. (2016) and Knopf et al. (2018). However, the aerosol chamber DMPS and cloud chamber SMPS aerosol size distributions do not indicate major growth of particles between transfer and cloud evacuation. A different explanation for our observations could be that larger heptadecane particles were present (and similarly larger dust particles in the dust experiment) before the evacuation started. Such particles could have swollen or activated into cloud particles sooner than the main mode. The SMPS size distributions only extend to 615 nm, thus, we cannot say whether larger particles were indeed present. Another explanation that cannot be ruled out completely is, that other aerosol particles from background contamination are activated and cause the first activation mode. However, it is not clear why contamination should only be existent in the heptadecane experiments and not in the $\alpha$-pinene and TMB experiments as well.

As Mikhailov et al. (2009) point out, organic (semi-)solid amorphous particles can kinetically limit the water uptake and may thus influence the growth activation as cloud condensation nuclei. Thus, the phase state of the aerosol particles (represented by their bounciness) could play a role in the onset of activation, as hinted in measurements by Ignatius et al. (2016); Ladino et al. (2014); Wagner et al. (2014). As the cloud chamber is generally at close to water

saturation at the beginning of each expansion, an earlier activation to cloud particles would be expected than seen here. It is likely, that the observation of cloud particles is delayed due to wall heating effects, i.e. the cloud forms in the middle of the chamber but when starting pumping air into the sample lines to the cloud particle instrumentation, air from the boundary layer between wall and chamber interior is drawn into the lines first which might be at slightly higher temperatures and therefore at lower relative humidity and cloud free. Such wall effects have been observed in other cloud chambers as the AIDA chamber as well (Möhler et al., 2003).

The comparison of activated fractions shows some differences between the investigated aerosol particles: while most experiments show no or only little signs of cloud processing in terms of a changed activated fraction of aerosol particles to cloud particles, TMB activated fractions in the second run are lower than in the first run. Thus, after the first cloud cycle the TMB particles are less likely to form cloud droplets, though the mean mode diameter shifted slightly to larger sizes which should foster faster droplet activation. $\alpha$-pinene shows a similar behaviour, though less obvious. Generally, cloud processing is thought to increase the efficiency of activation into cloud particles (e.g. Hoose et al., 2008), through changing the internal chemical structure and/or composition of the aerosol particles. However, as the experiments reported here exhibit pure SOA aerosol, and we expect no other organic and inorganic material (or vapours) in the chambers, cloud processing here will only change the aerosol mass, not aerosol chemistry. Only remaining organic vapour can condense into the droplets. Uptake of organic vapours during the first cloud cycle and thus, less vapour available during the second cloud cycle could lead to a smaller effect of co-condensation (Topping et al., 2013) and thus, smaller cloud particle numbers and reduced activated fractions. Dust as well showed a higher activated fraction during the first run, however, one has to bear in mind the higher pump speed used in that run. Heptadecane shows a contrary behaviour to the other SOA compounds: In the slow pump speed experiment, the second cloud cycle exhibits higher activated fractions than the first cycle with numbers comparable to those in the cloud cycles of the fast pump speed experiment. In the latter no significant difference in activated fractions between the two cloud runs can be distinguished. It has to be noted that the mean mode diameter of the aerosol in the runs using the fast pump speed were about 400nm, but approximately 500nm in the slow runs. In general, the heptadecane runs show significantly smaller activated fractions than the runs using the compounds in semi-solid phase state, even though the aerosol mean mode diameters are larger than in the TMB and $\alpha$-pinene experiments. Therefore, we speculate that the phase state of the SOA particles shows an impact on cloud activation here.

Pre-cooling of SOA particles has the potential to increase the aerosol particles' ability to act as INP (e.g. Ladino et al., 2014; Wagner et al., 2014). Thus, there could have been a higher chance of ice formation in the respective second cloud activation runs. However, apparently the minimum temperatures reached during the first cloud runs here were not cold enough for such a pre-activation of the aerosol particles and hence, no ice had formed.

A further small difference between the compounds in semi-solid and liquid phase state is the growth of particle sizes during cloud activation runs: While in the heptadecane runs (liquid) the MVD increases slightly with time, it stays fairly constant in the $\alpha$-pinene and TMB runs. This might be due to the smaller activated fractions in the heptadecane experiments which leave more water vapour for further growth of the particles further into the cloud evacuation.

## 5   Conclusions

The coupled system of the Manchester Aerosol and Ice Cloud Chamber has been used to investigate the ice nucleating ability of SOA particles at temperatures and relative humidities that are relevant to mixed-phase clouds. SOA particles were formed on precursors in the aerosol chamber by photo-oxidation. Clouds were formed by evacuation of the cloud chamber that led to a quasi-adiabatic drop in temperature from approximately -20°C to about -28°C/-25.5°C (fast/slow pump speed, respectively) fostering cloud formation. At the start of the chamber evacuation the humidity inside the chamber was close to water saturation, allowing for a speedy onset of cloud formation. The measurements show that the photo-oxidised SOA particles are not efficient ice nucleating particles in the tested temperature range: No ice formation was observed, irrespective of the type of SOA particles that were used (from $\alpha$-pinene, heptadecane, and TMB precursors), resembling biogenic/anthropogenic and semi-solid/liquid compounds. A sensitivity experiment using kaolinite showed that ice formation was possible with the given setup.

While the phase state (which is represented by the particles bounciness, see Sect. 3.1) of the particles has no measurable impact on ice nucleation under the reported conditions, the SOA particles of different phase state show differences in activation and cloud processing. The semi-solid SOA particles from TMB and $\alpha$-pinene precursors show (slightly) reduced activated fractions in a subsequent cloud cycle, the liquid SOA particles from heptadecane precursor reveals increased activated fractions. The exact reasons can only be speculated on as for example no measurements of organic vapours in the chambers are available. Furthermore, the heptadecane experiments show a two-fold cloud activation feature that is absent in the TMB and $\alpha$-pinene experiments. Again, these cannot be fully explained here as measurements which would be able to support or disapprove the speculations are missing.

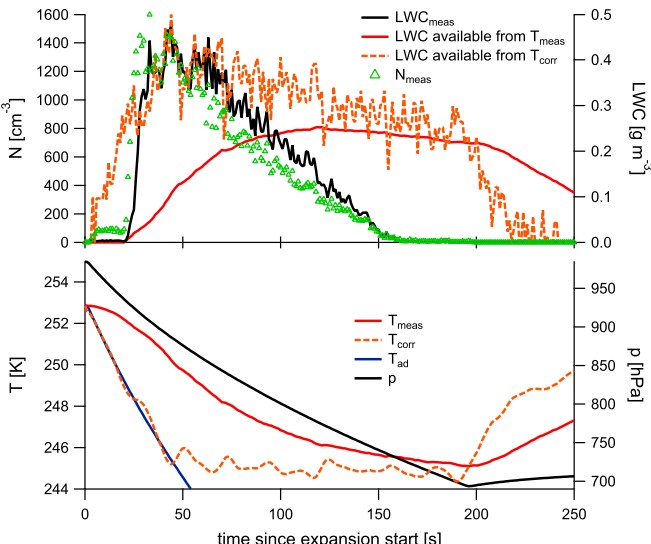

**Figure A1.** Measured and corrected temperatures and LWCs during the second heptadecane run of experiment 8 (cf. Table 1). See text for details.

The main conclusions from these experiments are that the tested photo-oxidised SOA particles do not nucleate ice under the mixed-phase cloud regime, neither in depositional nor immersion freezing mode. Thus, even in their high abundance in nature, SOA particles will act as cloud condensation nuclei and only as ice nucleating particles when cold enough (likely below the homogeneous freezing threshold).

## 6 Data availability

As the chambers are part of the EUROCHAMP consortium, the data will be made available at the EUROCHAMP data centre. Until then, they will be distributed upon request.

## Appendix A: Temperature correction

During evacuations the temperature change in the cloud chamber is quasi-adiabatic, if no clouds form. By considering the time constant, a quasi-adiabatic temperature drop can be seen at the beginning of the expansion, while heating effects become stronger later on from wall heating and latent heat release from droplet formation. Fig. A1 shows the temperatures during a cloud expansion (in the lower panel): measured temperature in red, calculated adiabatic temperature in blue, and the corrected temperature considering the time constant in orange and dashed. The corrected temperature is smoothed with a 20 seconds running mean due to the small scale fluctuations in the temperature measurements that otherwise propagate into the corrected temperature. Additionally, the plot shows the pressure. In the upper panel

the measured cloud particle number concentrations are displayed in green triangles. The sharp increase in cloud particle numbers at 20 seconds (i.e. the major activation) coincides with the departure in the corrected temperature curve from the adiabatic curve. By calculating the theoretically available humidity from these retrieved temperatures we find a good match with our LWC observations, as the calculated available LWCs in the upper panel show.

*Author contributions.* WF and DH main measurements operations and (preliminary - DH) data analysis, with help of JD and RA. Concept GM and PC. WF main writing. All contributed to discussions and fine tuning of manuscript.

*Competing interests.* The authors declare no competing interests.

*Acknowledgements.* The authors like to thank Lee Paul and Barry Gale for technical help with the chambers and the air system. The experiments reported in this manuscript were carried out in the project CCN-Vol_IN_Bounce that was funded by the Natural Environment Research Council (NERC) under the grant code NE/L007827/1. PC acknowledges funding from the European Union's 7th Framework Programme BACCHUS (FP7/2007-2013) under grant agreement number 603445. The MAC received funding from the European Union's Framework 7 EUROCHAMP2 Network and MAC and MICC both currently receive funding from the Horizon 2020 research and innovation programme through the EUROCHAMP-2020 Infrastructure Activity under grant agreement no. 730997. AV acknowledges fiunding from the European Research Council (ERC-StG QAPPA, 335478).

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
