# Peer review of "The efficiency of secondary organic aerosol particles to act as ice nucleating particles at mixed-phase cloud conditions"

_Atmospheric Chemistry and Physics, 2017_

## Referee Comment (RC1) · Anonymous Referee #1 · 30 Jan 2018

This manuscript examines biogenic and anthropogenic SOA surrogate particles for their ice nucleating ability. The SOA particles were photochemically generated in the Manchester aerosol chamber (MAS) and then transferred to the Manchester Aerosol and Ice Cloud Chamber (MICC) where ice nucleation was probed between -20 C to -28.6 C at water saturation mimicking mixed-phase cloud formation conditions. Reference ice nucleation experiments employing ammonium sulfate and kaolinite particles were conducted. Under probed conditions only kaolinite particles initiated ice nucleation.

The authors present a study of increasing interest, i.e. if and how organic, in particu-

lar, secondary organic aerosol (SOA) particles form ice in the atmosphere. This is an important topic and I am in support that new experimental results should be published. However, I find that this manuscript lacks discussion of recent literature on ice nucleation and diffusion of SOA particles to set the new results in the right context. SOA ice nucleation or diffusion has been studied by several groups in recent years (among others, Wang et al., 2012, Ignatius et al. 2016, Mohler et al., 2008, Charnawskas et al., 2017, Price et al., 2015, Wagner et al., 2017, Lienhard et al., 2015, Kanji et al., 2017, Ladino et al., 2014, recent review by Knopf et al., 2018). These papers should be present in introduction and may be further discussed in other sections of the manuscript.

Furthermore, the SOA generation procedures may vary among this and other studies. This should be mentioned/discussed in places.

I find the supplemental material should be better implemented within the main text. As is, there are some notes to it, but the supplement has a lot of important information. I feel the bounce experiments would be better situated in the main text, also to be more visible, but I leave this to the authors.

The figures in text and supplement reporting ice nucleation experiments should also include the supersaturation of ice, Sice. This is crucial information missing.

I recommend that the abstract states explicitly the particle systems investigated for ice nucleation

p. 2, l. 10-15: other studies mentioned above in general comment should be mentioned.

p. 3, l. 24-25: The 33 m transfer line. You show data later but please elaborate on particle losses due to diffusion, gravitational settling etc. What is the flow speed and pressure in this transfer line? Since this is a new experiment, it would be beneficial to know these parameters.

[Figure]

p. 5, l. 4-8: Here, I would give the bounce experiments more exposure. "Bouncy" is not really a physical parameter, is it possible to use phase state definitions, such as semi-solid, solid etc.?

p. 5, l. 19: Here, the reader learns the first time the NOx is involved in SOA formation. This can be different from above mentioned studies. What does this mean for SOA composition, viscosity etc.? This could be important but is not discussed.

p. 5, l. 26 – p. 6, l. 8: Some details are not entirely clear to me: The SOA from MAC flows into evacuated MICC. Then MICC is filled with gas. Do you expect losing SOA species due to evaporation (low pressure) and due to dilution? The VOCs then diffuse to the cold walls of MICC? Also going from a warm (MAC) to a cold environment (MICC), does this not affect RH fields, thus affecting organic phase state?

p. 6, l. 24-25: The air from MAC was humid and entered MICC. Are the particle RH trajectories known for the transfer? Does this impact phase state? See, e.g. discussion in Ignatius et al. (2016) and Knopf et al. (2018). The humid air condenses onto cold MICC walls?

p. 6, l. 30-31: Here and Fig. 4 case: Are activated droplet sizes what would be expected from Kohler theory and diffusional growth?

p. 7, l. 27-35: For this discussion it is crucial to know also the temperature and Sice values during measurement of the activated fraction. At this point the discussion is confusing and one wonders about these results. Maybe at fast pumping speed, i.e. at high Sice, the activated fraction of HD is not as sensitive compared to lower pumping speed and thus lower Sice?

p. 8, l. 16-21: Is it possible to make this speculative discussion a bit more quantitative?

p. 8, l. 29-31: I find this too simplified and feel it needs more discussion. Please look at studies mentioned above in general comment.

p. 9, l. 1-5: I find this needs more discussion. Please look at studies mentioned

above in general comment. SOA particles are produced in different ways, different temperatures are probed etc.

p. 9, l. 12-13: Different heptadecane properties due to different VOC/NOx ratio applied?

p. 9, l. 14-15: Not necessarily higher supersaturation are needed but longer times. The need of larger supersaturation may be "apparent", i.e. disequilibrium between gas and condensed phase.

p. 9, l. 15-16: Indeed, different temperature in chambers or sampling lines can affect particle properties. See e.g. discussion in Knopf et al. (2018) and Ignatius et al. (2016).

p. 9, l. 28-30: Above mentioned literature may enhance this discussion.

p. 10, l. 5-6: Please refer to, e.g., Wagner et al. (2014, 2012) articles.

p. 10, l. 16-18: Heptadecane is more viscous and therefore the activated fraction is lower?

Figures: 3-5 and similar ones in supplement: specific for figures in main text: particle images are not described in caption. Panel indicators are missing. As stated above, please include Sice. It is confusing to have a legend in third panel that includes definitions for other panels. Please split legend to corresponding panels.

Figure 5: What do you mean by "eating up"? A Wegener-Bergeron-Findeisen process? Please change expression.

Table 2: You mean MAC and not "aerosol chamber"? "Amount injected": unit? Please elaborate. Have aqueous solutions been injected? What is the mass? The mole fraction or other information is needed. Units missing for gas species. The mass difference, last column, between pinene and TMB is correct?

Table 3: How can mass in MICC be larger than in MAC (mass_DMPS vs.

mass_SPMS)?

Supplement: p. 1: change "cooking" to "processing" or other terminology.

Figure S4: Please include pinene SOA bounce fraction.

p. 4: Change "ingredients" to "species" or "compounds", etc.

What is the difference between S17 and S19 and S18 and S20 experiments? Maybe additional text is necessary?

Technical corrections:

I suggest throughout manuscript and supplement to change the expression "cloud evacuation" to "cloud activation experiment" or something along those lines.

p. 7, l. 25: Change language. Avoid the term "sister run".

p. 8, l. 26: Maybe use "employed" or "applied" instead of "used".

p. 9, l. 8: Avoid "kicks in". Change language.

p. 9, l. 30: Exchange "than" with "as".

p. 9, l. 32: Instead of "sucking" use" pumping" or "evacuating".

p. 10, l. 6-7: It feels there is an error in this sentence.

p. 10, l. 30: Exchange "no" for "not".

p. 11, l. 22: Exchange "bump" to "maximum" or similar.

References:

Ignatius, K., Kristensen, T. B., Jarvinen, E., Nichman, L., Fuchs, C., Gordon, H., Herenz, P., Hoyle, C. R., Duplissy, J., Garimella, S., Dias, A., Frege, C., Hoppel, N., Troestl, J., Wagner, R., Yan, C., Amorim, A., Baltensperger, U., Curtius, J., Donahue, N. M., Gallagher, M. W., Kirkby, J., Kulmala, M., MoÌĹhler, O., Saathoff, H., Schnaiter, M.,

Tome, A., Virtanen, A., Worsnop, D., and Stratmann, F.: Heterogeneous ice nucleation of viscous secondary organic aerosol produced from ozonolysis of $\alpha$-pinene. Atmos. Chem. Phys., 16, 6495−6509, 2016.

Knopf, D. A., Alpert, P. A., and Wang, B.: The Role of Organic Aerosol in Atmospheric Ice Nucleation: A Review, ACS Earth and Space Chemistry, DOI: 10.1021/acsearthspacechem.7b00120, 2018.

Mohler, O., Benz, S., Saathoff, H., Schnaiter, M., Wagner, R., Schneider, J., Walter, S., Ebert, V., and Wagner, S.: The effect of organic coating on the heterogeneous ice nucleation efficiency of mineral dust aerosols. Environ. Res. Lett., 3, 025007, 2008.

Wang, B. B., Lambe, A. T., Massoli, P., Onasch, T. B., Davidovits, P., Worsnop, D. R., and Knopf, D. A.: The deposition ice nucleation and immersion freezing potential of amorphous secondary organic aerosol: Pathways for ice and mixed-phase cloud formation. J. 2188 Geophys. Res., 117, D16209, 2012.

Price, H. C., Mattsson, J., Zhang, Y., Bertram, A. K., Davies, J. F., Grayson, J. W., Martin, S. T., O'Sullivan, D., Reid, J. P., Rickards, A. M. J., and Murray, B. J.: Water diffusion in atmospherically relevant $\alpha$-pinene secondary organic material. Chem. Sci., 6, 4876−4883, 2015.

Kanji, Z. A., Ladino, L. A., Wex, H., Boose, Y., Burkert-Kohn, M., Cziczo, D. J., and Kramer, M.: Overview of Ice Nucleating Particles. In Ice Formation and Evolution in Clouds and Precipitation: Measurement and Modeling Challenges; American Meteorological Society, 58, pp 1.1−1.33, 2017.

Wagner, R., Ho̍L̍hler, K., Huang, W., Kiselev, A., Mohler, O., Mohr, C., Pajunoja, A., Saathoff, H., Schiebel, T., Shen, X., and Virtanen, A.: Heterogeneous ice nucleation of $\alpha$-pinene SOA particles before and after ice cloud processing. J. Geophys. Res., 122, 4924−4943, 2017.

Lienhard, D. M., Huisman, A. J., Krieger, U. K., Rudich, Y., Marcolli, C., Luo, B. P.,

Bones, D. L., Reid, J. P., Lambe, A. T., Canagaratna, M. R., Davidovits, P., Onasch, T. B., Worsnop, D. R., Steimer, S. S., Koop, T., and Peter, T.: Viscous organic aerosol particles in the upper troposphere: diffusivity-controlled water uptake and ice nucleation? Atmos. Chem. Phys., 15, 13599−13613, 2015.

Wagner, R., Mohler, O., Saathoff, H., Schnaiter, M., Skrotzki, J., Leisner, T., Wilson, T. W., Malkin, T. L., and Murray, B. J.: Ice cloud processing of ultra-viscous/glassy aerosol particles leads to enhanced ice nucleation ability. Atmos. Chem. Phys., 12 (18), 8589−8610, 2012.

Wagner, R., Mohler, O., Saathoff, H., and Schnaiter, M.: Enhanced high-temperature ice nucleation ability of crystallized aerosol particles after preactivation at low temperature. J. Geophys. Res., 119, 19, 2014.

Ladino, L. A., Zhou, S., Yakobi-Hancock, J. D., Aljawhary, D., Abbatt, J. P. D. Factors controlling the ice nucleating abilities of $\alpha$-pinene SOA particles. J. Geophys. Res., 119, 9041−9051, 2014.

---

## Referee Comment (RC2) · Anonymous Referee #2 · 7 Feb 2018

The manuscript by Frey and co-workers investigates the ice nucleation ability of secondary organic aerosol (SOA) particles under so-called mixed-phase cloud conditions. SOA particles have been found to nucleate ice under so-called cold cloud conditions found at high atmospheric altitudes, an ability which is often ascribed to their glassy solid phase state, but their ability for cloud formation at much higher temperatures / lower altitudes has not yet been proven. This work is well-written and addresses this very relevant problem by connecting an aerosol formation vessel (Manchester Aerosol Chamber, MAC) and a vessel for probing the ice nucleation ability of particles (Manchester Ice Cloud Chamber, MICC) in an innovative experimental setup. The authors tested an array of different SOA particles by using three different precursors: one from

biogenic and two from anthropogenic origin, making the work relevant for the ambient atmosphere and hence publication in ACP. The main result of this work is that none of the investigated organic aerosols facilitated ice nucleation in the observed temperature and RH region. Such negative results are important to direct future works on the relevance of these particles for cloud formation. However, in my understanding, there was no possibility that ice nucleation should be observed due to the way the experiments were operated. This greatly diminishes the value of this publication and might disincentivize further research on the topic if it won't be stressed clearly by the authors why ice nucleation did not occur. I would like to outline my thoughts in the following and already encourage the author's to disagree with me in their rebuttal in case I misunderstood the experimental setup. At this point, however, I cannot recommend the manuscript for publication.

**General critique**

In the following I will assume that a glassy phase state is necessary for SOA particles to nucleate ice. If this is the case, a look at the phase diagram of glassy organic material should have sufficed to predict the lacking ice nucleation ability of the SOA particles observed in these experiments (e.g. Koop et al., 2011). The authors state that the MICC was at water saturation before expansion and during the filling of the vessel. At water saturation, a hygroscopic substance cannot be in a viscous or glassy phase state, no matter the glass transition temperature of the pure substance. In previous chamber experiments showing the ice nucleation ability of organic aerosols (e.g. Murray et al. (2010), Wilson et al. (2012)), expansion was always initialized at around ice saturation, thus far below water saturation level. The only reason particles could be in a viscous state in the experiment at hand would be if they would cool faster than they could take up water upon entering MICC and hence maintain a non-equilibrium state. These dynamics however seem very difficult to predict since the exact temperature and humidity profiles between MAC and MICC are probably difficult to obtain. Hence, the authors investigated in this study not only if organic aerosols could induce ice nucleation at

higher temperatures than expected, but also when initialized at higher humidities than in previous studies. Ice nucleation by organic aerosols under mixed-phase cloud conditions should only be possible if the glassy state is somehow maintained despite the high temperatures and humidities. In general, the authors miss to mention that not only temperature is important to determine aerosol phase state, but also humidity. A study by Berkemeier et al. (2014) investigating the kinetics of water uptake of viscous organic particles shows that there is a sensitive interplay between temperature, initial humidity and humidification rate that enables organic aerosols to be in the glassy state at humidities relevant for ice nucleation. If it was not the goal to use the glassy phase state of organic aerosols to trigger ice nucleation this has to be stated more clearly in this manuscript. In this case however, the discussion devoted to bounciness of particles would be superfluous (or even misleading) and a different incentive to discuss organic aerosols as ice nuclei would have to be presented. It is hard to imagine ice nucleation on liquid aerosol particles.

**Specific remarks**

- In the introduction, it seems worth mentioning that many more studies have investigated ice formation from organic aerosols. Examples here are works from the Knopf group (e.g. Wang et al., 2012; Charnawskas et al., 2017), the Tolbert group (Baustian et al., 2013) or at the AIDA chamber (e.g. Wilson et al., 2012). Overall the number of references in this paper is very lacking and too much focused on the author's own work.

- As mentioned above, the introduction too much focused on temperatures, too little on relative humidities. The biggest problem with ice nucleation and glassy SOA are the high relative humidities necessary to nucleate ice on the particles, which in turn leads to non-viscous phase states. A discussion on this competition process can be found in Berkemeier et al. (2014).

- The usage of the words "bouncy" and "non-bouncy" in this study are misleading. What the author's should say are the conditions under which these particles bounce (RH and T). Do they bounce at -20 °C and water saturation? Hence, in the context of p.8, l.23, this is not a meaningful statement.

• The effects of cloud processing in the TMB vs heptadecane SOA experiment are an interesting finding. Is the picture of TMB SOA showing lower activated fractions in the second chamber evacuation run consistent among repetitions of the experiment?

**Minor and technical comments**

• p. 4, l.2 – boiling point of water

• p. 4, l.11 – Out of curiosity, why would the oxidation be conducted with ozone and not also under irradiation with UV light to generate OH radicals? Most products of SOA formation should not be susceptible to reaction with ozone.

• p.5, l.11 – A closing parenthesis ")" is missing here.

• p.5, l.17 – It is not entirely clear what is meant with "to achieve this". I assume it refers to the cleaning of the chamber, not the particle formation described in the previous sentence.

• p. 7, l.6 – I believe the word "one" is superfluous here.

• p.9, l.12 – Not the precursors are bouncy, the SOA from these precursors are, at a specific RH and T.

• p. 9, l.13 – The statement confuses kinetics and thermodynamics, please clarify that not a higher supersaturation is needed to activate viscous particles, it just takes longer time.

**References**

Baustian, K. J. et al. State transformations and ice nucleation in amorphous (semi-)solid organic aerosol. Atmos. Chem. Phys. 13, 5615–5628 (2013).

Berkemeier, T., Shiraiwa, M., Pöschl, U. Koop, T. Competition between water uptake and ice nucleation by glassy organic aerosol particles. Atmos. Chem. Phys. 14, 12513–12531 (2014).

Charnawskas, J. C. et al. Condensed-phase biogenic–anthropogenic interactions with implications for cold cloud formation. Faraday Discuss. 200, 165–194 (2017).

Koop, T., Bookhold, J., Shiraiwa, M. Pöschl, U. Glass transition and phase state of organic compounds: dependency on molecular properties and implications for secondary organic aerosols in the atmosphere. Phys. Chem. Chem. Phys. 13, 19238 (2011).

Murray, B. J. et al. Heterogeneous nucleation of ice particles on glassy aerosols under cirrus conditions. Nat. Geosci. 3, 233–237 (2010).

Wang, B. et al. The deposition ice nucleation and immersion freezing potential of amorphous secondary organic aerosol: Pathways for ice and mixed-phase cloud formation. J. Geophys. Res. 117, (2012).

Wilson, T. W. et al. Glassy aerosols with a range of compositions nucleate ice heterogeneously at cirrus temperatures. Atmos. Chem. Phys. 12, 8611–8632 (2012).

---

## Author Comment (AC1) · 11 May 2018

Reply to Referee 1:

We'd like to thank the reviewer for her/his comments which helped to greatly improve the manuscript. In the following, we provide replies to the reviewer's comments (typeset in bold).

**This manuscript examines biogenic and anthropogenic SOA surrogate particles for their ice nucleating ability. The SOA particles were photochemically generated in the Manchester aerosol chamber (MAS) and then transferred to the Manchester Aerosol and Ice Cloud Chamber (MICC) where ice nucleation was probed between -20 C to -28.6 C at water saturation mimicking mixed-phase cloud formation conditions. Reference ice nucleation experiments employing ammonium sulfate and kaolinite particles were conducted. Under probed conditions only kaolinite particles initiated ice nucleation. The authors present a study of increasing interest, i.e. if and how organic, in particular, secondary organic aerosol (SOA) particles form ice in the atmosphere. This is an important topic and I am in support that new experimental results should be published.**
We are happy about and thank the reviewer for this positive assessment.

**However, I find that this manuscript lacks discussion of recent literature on ice nucleation and diffusion of SOA particles to set the new results in the right context. SOA ice nucleation or diffusion has been studied by several groups in recent years (among others, Wang et al., 2012, Ignatius et al. 2016, Mohler et al., 2008, Charnawskas et al., 2017, Price et al., 2015, Wagner et al., 2017, Lienhard et al., 2015, Kanji et al., 2017, Ladino et al., 2014, recent review by Knopf et al., 2018). These papers should be present in introduction and may be further discussed in other sections of the manuscript.**
We gave the introduction a major overhaul and also revised the discussion to include more references and compare to the referenced works.

**Furthermore, the SOA generation procedures may vary among this and other studies. This should be mentioned/discussed in places.**
We agree that the SOA generation procedures can vary substantially among different studies. We mentioned this fact and included mentioning/discussion in the respective sections (i.e. mainly introduction and discussion).

**I find the supplemental material should be better implemented within the main text. As is, there are some notes to it, but the supplement has a lot of important information. I feel the bounce experiments would be better situated in the main text, also to be more visible, but I leave this to the authors.**
Following the reviewers suggestion, we moved the bounce experiments into the main text as new Section 3.1 and slightly adapted the text due to its new position.

**The figures in text and supplement reporting ice nucleation experiments should also include the supersaturation of ice, Sice. This is crucial information missing.**
Unfortunately, the TDL system was malfunctioning during the experiments, thus only the CR4 provided humidity measurements. However, the CR4 can only sample at ambient pressure, which means, there are no humidity measurements during cloud activation runs. We provide initial $S_{ice}$ (i.e. measured just before a cloud activation run) for each cloud activation run. Due to the phase changes from water vapour into the particle phase, simulation of $S_{ice}$ would yield high uncertainties; therefore, we do not provide these. However, we included the initial $S_{ice}$ measurements in the overview in Table 1.

**I recommend that the abstract states explicitly the particle systems investigated for ice nucleation.**
We added: "These are namely alpha-pinene, heptadecane, and 1,3,5-trimethylbenzene."

**p. 2, l. 10-15: other studies mentioned above in general comment should be mentioned.**
We have rewritten the introduction, and now it includes references to other studies as the ones you have mentioned above.

**p. 3, l. 24-25: The 33 m transfer line. You show data later but please elaborate on particle losses due to diffusion, gravitational settling etc. What is the flow speed and pressure in this transfer line? Since this is a new experiment, it would be beneficial to know these parameters.**
The flow speed and pressure in the transfer line are governed by the pressure (difference) in MICC and MAC. At the beginning of a transfer, the pressure in the transfer line close to MAC will be close to the pressure in MAC, while at the end close to MICC, it will be rather at MICC pressure. As the pressure difference reduces over time (when MICC is filling) the flow speed decreases. A typical flow speed time series for a refill from MAC after a MICC cloud activation is shown below.

[Figure]

Figure: Volume flow in transfer pipe during refill of MICC from MAC after a cloud activation.

The comparison between DMPS size distribution measurements from MAC just before transfer and SMPS size distributions measured in MICC just after the transfer can be used to infer how severe particle losses during transfer are: The figure below shows the respective size distributions for the different SOA experiments. In case of the alpha-pinene SOA, no significant change in the main particle mode diameter is apparent, while for the first heptadecane experiment and the TMB experiment growth in the main particle mode happened, possibly in MICC due to the colder temperatures there compared to Mac temperature which fosters the condensation of the organic vapours onto the particles.

[Figure]

Figure: Comparison of size distributions just before transfer in MAC and just after transfer in MICC.

**p. 5, l. 4-8: Here, I would give the bounce experiments more exposure. "Bouncy" is not really a physical parameter, is it possible to use phase state definitions, such as semi-solid, solid etc.?**

We moved the section about the particle bounce measurements into the main manuscript at the end of the introduction part of section 3 and modified it slightly to adapt to the new position.

We agree with the reviewer regarding the phase state definitions and use phase state definitions (semisolid or liquid) instead.

**p. 5, l. 19: Here, the reader learns the first time the NOx is involved in SOA formation. This can be different from above mentioned studies. What does this mean for SOA composition, viscosity etc.? This could be important but is not discussed.**

The effect of $NO_x$ on SOA viscosity is not studied (at least the authors are not aware of any publications on this). It should be noted though, that also the O:C of produced SOA was relatively low in these experiments. As shown in Pajunoja et al. (2015) the particle viscosity (or bounce behaviour) at elevated RHs depends strongly on O:C. This is because the particle hygroscopicity (i.e. the particle phase water content at certain humidity) increases with O:C. Hence, not only the $NO_x$ might affect the composition or viscosity, but also the oxidation conditions more generally.

The use of $NO_x$ in these experiments was to help initiate the photochemistry, and the work did not include a systematic investigation of the effect of $NO_x$ on particle composition or properties.

We added this discussion to the manuscript.

**p. 5, l. 26 – p. 6, l. 8: Some details are not entirely clear to me: The SOA from MAC flows into evacuated MICC. Then MICC is filled with gas. Do you expect losing**

**SOA species due to evaporation (low pressure) and due to dilution? The VOCs then diffuse to the cold walls of MICC? Also going from a warm (MAC) to a cold environment (MICC), does this not affect RH fields, thus affecting organic phase state?**

Correct, the evacuated MICC is refilled from MAC until ambient pressure is reached. Thus, there is no further dilution of the aerosol population other than by the air that remained in MICC prior to the transfer (it is not possible to evacuate MICC to vacuum, but the lowest pressure is 200hPa). As stated in the text, about $8m^2$ air from MAC (which holds the aerosol) is diluted by the remaining $2m^2$ of filtered air in MICC. The dilution leads to a tendency for components to evaporate. The transferred air cools in MICC and cooling leads to a tendency for the components to condense. Thus, there are opposing tendencies and any semi-volatile component in the aerosol will have a tendency to transfer between phases accordingly. The rate at which the components follow this tendency may possibly be influenced by the changes in condensed phase properties with temperature, since an increase in viscosity towards or across the glass transition may lead to a decrease in condensed phase diffusion and decrease in rate of bulk equilibration.

Effects of dilution and cooling can be investigated to a certain amount by comparing the SMPS and CPC measurements from directly following the transfer of air from MAC to MICC (when MICC temperatures were above the target temperature) and the repeated measurements once the target temperature was reached. For example, looking at the SMPS size distributions given in Figures 3 and 4 (and others in the supplement), which shows the size distribution directly "after transfer" in red and the two size distributions sampled when the target temperature was reached "S1/S2 before evacuation" in blue colours, no significant changes are apparent. Thus, it seems that enough organic vapour remained in the gas phase in MAC prior to transfer such that particles would not evaporate/shrink when entering MICC even if amounts of the vapour are lost to condensation on the chamber walls.

We added some explanation to the manuscript.

**p. 6, l. 24-25: The air from MAC was humid and entered MICC. Are the particle RH trajectories known for the transfer? Does this impact phase state? See, e.g. discussion in Ignatius et al. (2016) and Knopf et al. (2018). The humid air condenses onto cold MICC walls?**

The exact RH trajectories are not known. We do have information about the initial RH in MAC and the RH in MICC after transfer. Due to the not-ambient pressure it is not possible to measure the RH during transfer.

MICC walls are ice coated from the humidity that entered the chamber during the cleaning cycles. Thus, any humidity in excess of $Rh_{ice} = 100\%$ would condense onto the walls after the transfer. However, this process will take some time, following a rough calculation assuming a gradient in saturation ratio of 0.15 over 0.5m (half the diameter of MICC), molecular diffusion of water vapour is really small and the supersaturation would stay there for maybe 1.5 hours. There will be turbulence in the chamber too (from the air stream filling the chamber), which should reduce this time, and the presence of particles (that have their own water content, eventually subsaturated wrt liquid water) will prolong the diffusion, too. Generally, RH can have an impact on the phase state of the SOA particles. From the bounce measurements presented here, we can estimate that even at low RH (and thus at high RH as well) the SOA particles from heptadecane precursor will be liquid, while TMB is in a semi-solid phase even at high RH.

**p. 6, l. 30-31: Here and Fig. 4 case: Are activated droplet sizes what would be expected from Kohler theory and diffusional growth?**

Generally, yes! However, the size distribution will be broader than theory predicts due to non-uniformities in the temperature and humidity fields in the chamber (see also reply to comment "p. 8, l. 16-21" below).

**p. 7, l. 27-35: For this discussion it is crucial to know also the temperature and $S_{ice}$ values during measurement of the activated fraction. At this point the discussion is confusing and one wonders about these results. Maybe at fast pumping speed, i.e. at high $S_{ice}$, the activated fraction of HD is not as sensitive compared to lower pumping speed and thus lower $S_{ice}$?**

We modified Figure 6 which now also includes the temperature measurements during the cloud expansions and the initial $S_{ice}$. Due to the failure of the TDL system, we unfortunately, only have humidity measurements at ambient pressure, thus, not during the cloud expansion experiments. We included statements about the temperatures and initial $S_{ice}$ in the manuscript.

"In the slower pump speed experiment, however, the second heptadecane run shows a higher activated fraction than the first run, though initial ice supersaturation are the same and tepmeratures in both runs are within 0.2°C. Thus, the aerosol becomes more efficient at activating to droplets. The first run here exhibits lower activated fractions as the fast pump speed runs, the second run peak activated fraction is about the same as in the fast pump speed runs. The α-pinene slow pump speed experiment shows the opposite behaviour, the second cloud run has slightly lower activated fractions as the first. The same is true for the TMB slow pump speed runs. However, the initial temperatures differ by about 0.7°C resulting in a less strong temperature drop during expansion, and also the initial ice supersaturation differs."

**p. 8, l. 16-21: Is it possible to make this speculative discussion a bit more quantitative?**

We expanded this discussion:

"... it will take time for the cloud to extend downwards to the bottom and be sampled, see also discussion in Möhler et al. (2003) for similar effects in the AIDA cloud chamber.

The observed cloud particle size distribution is wider than the simulated one. This could be due to effects of non-uniformity in temperature and humidity that lead to broadening of the size distribution. For example, as the chamber walls will stay at close to their initial temperature while the chamber centre cools adiabatically during an expansion, wall heating effects can create a temperature gradient within the chamber (warmer towards the walls and colder towards the chamber centre), which will induce a humidity gradient as well. The incoming air stream causes mixing which on the one hand side can reduce this gradient, but on the other side can induce inhomogeneous temperature and humidity fields leading to variation in the activation and growth of the cloud particles."

**p. 8, l. 29-31: I find this too simplified and feel it needs more discussion. Please look at studies mentioned above in general comment.**

We made significant changes to this section of the discussion, where we refer to other studies, comparing those results and conditions used during SOA generation with our experiments.

**p. 9, l. 1-5: I find this needs more discussion. Please look at studies mentioned above in general comment. SOA particles are produced in different ways, different temperatures are probed etc.**

We expanded the discussion and made significant changes, see comment above.

**p. 9, l. 12-13: Different heptadecane properties due to different VOC/NOx ratio applied?**

As mentioned above, the effect of $NO_x$ on SOA viscosity is not studied. It is difficult to speculate on this given the lack of any specific work on the issue. Also, given the relatively small difference in VOC/ $NO_x$ ratios in our experiments, it is probably unlikely that this would be the main reason behind the difference in the bounce behaviour.

**p. 9, l. 14-15: Not necessarily higher supersaturation are needed but longer times. The need of larger supersaturation may be "apparent", i.e. disequilibrium between gas and condensed phase.**

We rephrased:

"Thus, it takes a longer time for these viscous SOA particles to take on water vapour and grow due to diffusion limitations."

**p. 9, l. 15-16: Indeed, different temperature in chambers or sampling lines can affect particle properties. See e.g. discussion in Knopf et al. (2018) and Ignatius et al. (2016).**

True, we expanded the discussion by adding "Indeed, an effect of different temperatures in the chambers or sampling lines had been found and discussed by e.g. Ignatius et al. (2016) and Knopf et al. (2018)." to the end of this possible explanation.

**p. 9, l. 28-30: Above mentioned literature may enhance this discussion.**

We expanded "Thus, the phase state of the aerosol particles (represented by their bounciness) could play a role in the onset of activation, as hinted in measurements by e.g. Ignatius et al. (2016); Ladino et al. (2014); Wagner et al. (2014)."

Furthermore, the discussion has been modified substantially, and we have referenced and discussed above mentioned literature in various places throughout the discussion now.

**p. 10, l. 5-6: Please refer to, e.g., Wagner et al. (2014, 2012) articles.**

We added at the end of this passage:

"Pre-cooling of SOA particles has the potential to increase the aerosol particles' ability to act as INP (e.g. Ladino et al., 2014; Wagner et al., 2014). Thus, in the respective second cloud activation runs, there could have been a higher chance of ice formation. However, apparently the minimum temperatures reached during the first cloud runs here were not cold enough for such a pre-activation of the aerosol particles and hence, no ice had formed."

**p. 10, l. 16-18: Heptadecane is more viscous and therefore the activated fraction is lower?**

Actually, based on the bounce measurements heptadecane is less viscous than alpha-pinene and TMB (mentioned p9, l13, see also bounce figure in supplement or Saukko et al., 2012). The heptadecane SOA behaves as liquid droplets even at dry RH.

**Figures: 3-5 and similar ones in supplement: specific for figures in main text: particle images are not described in caption. Panel indicators are missing. As stated above, please include Sice. It is confusing to have a legend in third panel that includes definitions for other panels. Please split legend to corresponding panels.**

The figures have been altered as suggested. As mentioned before, we unfortunately can only state the initial $S_{ice}$ for these experiments. Due to the formation of cloud particles, i.e. phase change of vapour to particles, a calculation of $S_{ice}$ would be hampered by high uncertainties. The initial $S_{ice}$ is now included in Table 1 (alongside initial $S_{water}$).

**Figure 5: What do you mean by "eating up"? A Wegener-Bergeron-Findeisen process? Please change expression.**
We rephrased: "The data show formation of ice in a second mode, and the decrease and almost disappearance of particle numbers of smaller drops over time (Wegener-Bergeron-Findeisen process)."

**Table 2: You mean MAC and not "aerosol chamber"? "Amount injected": unit? Please elaborate. Have aqueous solutions been injected? What is the mass? The mole fraction or other information is needed. Units missing for gas species. The mass difference, last column, between pinene and TMB is correct?**
MAC is the synonym for the Manchester Aerosol Chamber, thus, both would be correct. The gas species are in $nmol\,mol^{-1}$, we added that information to the table. The VOCs were injected as high purity chemical liquids into a heated glass bulb where it immediately evaporates. A continuous flow of nitrogen then flushes the gas phase VOCs into the aerosol chamber. (See description in original manuscript p3 l7-9 and p5 l18/19.) We now expanded: "To prepare the system for injection of relevant gases for particle formation in MAC the volatile organic compound (VOC) injection glass bulb is heated and continuously flushed with nitrogen. The precursors for the SOA are injected into the glass bulb in form of high purity liquids, where they evaporate immediately. The vapourised VOCs and $NO_x$ are then injected during the last filling of the MAC air bag."
As we use pure liquids, their mole fraction is unity. To avoid confusion we changed the way we state the amount of VOCs injected: As we calculated the amount of liquid for the volume injection, taking into account the chamber bag volume, we now state the "nominal VOC mass" that should enter the chamber as gas. We also made a reference to the text, Sect. 3.1 Experimental Design.
Since we injected precursor gases with very different characteristics, the resulting aerosol differs not only by mass but also by e.g. their nucleation and growth characteristics and losses to the walls. TMB is much harder to produce and therefore, masses are smaller than for alpha-pinene, which contrary nucleates much faster. There was indeed an error in the TMB mass, but still the TMB mass is much smaller than e.g. the alpha-pinene mass.

**Table 3: How can mass in MICC be larger than in MAC (mass_DMPS vs. mass_SPMS)?**
Due to the differing size ranges of the DMPS and SMPS, the mass retrieved from these instruments can be different even if sampled from the same aerosol population. As the SMPS starts measuring at 13.8nm (vs 40nm DMPS) it will add mass from the small particles. If you look at the comparison of size distributions (figure above in the reply to your comment on the transfer pipe), in the α-pinene and TMB cases also larger particles are observed by the SMPS, which are absent in the DMPS measurements. These contribute to the SMPS mass accordingly.

**Supplement: p. 1: change "cooking" to "processing" or other terminology.**
Changed to: "...for any experiment performed on specified aerosol photochemically produced in the aerosol chamber."

**Figure S4: Please include pinene SOA bounce fraction.**
Since the bounce measurements shown here were not from the same experiments but from a previous experiment, there were unfortunately, no measurements of pure alpha-pinene SOA. However, for example Saukko et al. (2012) and Pajunoja et al. (2015) did show by flow tube measurements that alpha-pinene is semi-solid. We state more clearly now in the main text that the bounce measurements and the ice nucleation

measurements were not made in the same experiment but the bounce measurements were part of an earlier experiment using the same set up as our measurements.

**p. 4: Change "ingredients" to "species" or "compounds", etc.**
Changed to: "...all chemical substances as in a normal SOA experiment were used, without the actual precursor."

**What is the difference between S17 and S19 and S18 and S20 experiments? Maybe additional text is necessary?**
Yes, it seems additional text should be provided. There were two experiments with ammonium sulfate aerosol, with two cloud activation runs each. Thus, the difference between S17 and S19/S18 and S20 is the actual experiment, S17/S18 are cloud activation runs from experiment 2, S19/S20 from experiment 9 (see also Table 1 in main manuscript). We included a reference to the respective experiment (and the table) for all cloud activation run figures in the supplement.

**Technical corrections:**
**I suggest throughout manuscript and supplement to change the expression "cloud evacuation" to "cloud activation experiment" or something along those lines.**
We changed according to the reviewer's suggestion.

**p. 7, l. 25: Change language. Avoid the term "sister run".**
Changed to: "...the already shown heptadecane [...] and dust [...] runs accompanied by the respective other runs in the same experiments, plus a further heptadecane experiment...".

**p. 8, l. 26: Maybe use "employed" or "applied" instead of "used".**
Changed to "employed".

**p. 9, l. 8: Avoid "kicks in". Change language.**
Changed to "commences".

**p. 9, l. 30: Exchange "than" with "as".**
Done.

**p. 9, l. 32: Instead of "sucking" use" pumping" or "evacuating".**
Changed to "pumping".

**p. 10, l. 6-7: It feels there is an error in this sentence.**
It is not clear where the reviewer suspects an error. However, we added some information to the previous sentence and hopefully this will resolve the misunderstanding?
"Generally, cloud processing is thought to increase the efficiency of activation into cloud particles (e.g. Hoose et al., 2008), through changing the internal chemical structure and/or composition of the aerosol particles. However, as the experiments reported here exhibit pure SOA aerosol, and we expect no other organic and inorganic material (or vapours) in the chambers, cloud processing here will only change the aerosol mass, not aerosol chemistry."

**p. 10, l. 30: Exchange "no" for "not".**
Done.

**p. 11, l. 22: Exchange "bump" to "maximum" or similar.**
Rephrased sentence:
"The sharp increase in cloud particle numbers at 20 seconds (i.e. the major activation) coincides with the departure in the corrected temperature curve from the adiabatic curve."

*References:*
Hoose, C., Lohmann, U., Bennartz, R., Croft, B., and Lesins, G.: Global simulations of aerosol processing in clouds, Atmos. Chem. Phys., 8, 6939–6963, doi:10.5194/acp-8-6939-2008, 2008.

Ignatius, K., Kristensen, T. B., Järvinen, E., Nichman, L., Fuchs, C., Gordon, H., Herenz, P., Hoyle, C. R., Duplissy, J., Garimella, S., Dias, A., Frege, C., Höppel, N., Tröstl, J., Wagner, R., Yan, C., Amorim, A., Baltensperger, U., Curtius, J., Donahue, N. M., Gallagher, M. W., Kirkby, J., Kulmala, M., Möhler, O., Saathoff, H., Schnaiter, M., Tomé, A., Virtanen, A., Worsnop, D., and Stratmann, F.: Heterogeneous ice nucleation of viscous secondary organic aerosol produced from ozonolysis of α-pinene, Atmos. Chem. Phys., 16, 6495–6509, doi:10.5194/acp-16-6495-2016, 2016.

Knopf, D. A., Alpert, P. A., and Wang, B.: The Role of Organic Aerosol in Atmospheric Ice Nucleation: A Review, ACS Earth Space Chem., 2, 168-202, doi:10.1021/acsearthspacechem.7b00120, 2018.

Ladino, L. A., Zhou, S., Yakobi-Hancock, J. D., Aljawhary, D., and Abbatt, J. P. D.: Factors controlling the ice nucleating abilities of α-pinene SOA particles, J. Geophys. Res. Atmos., 119, 9041–9051, doi:10.1002/2014JD021578, 2014.

Möhler, O., Stetzer, O., Schaefers, S., Linke, C., Schnaiter, M., Tiede, R., Saathoff, H., Krämer, M., Mangold, A., Budz, P., Zink, P., Schreiner, J., Mauersberger, K., Haag, W., Kärcher, B., and Schurath, U.: Experimental investigation of homogeneous freezing of sulphuric acid particles in the aerosol chamber AIDA, Atmos. Chem. Phys., 3, 211–223, doi:10.5194/acp-3-211-2003, 2003.

Pajunoja, A., Lambe, A. T., Hakala, J., Rastak, N., Cummings, M. J., Brogan, J. F., Hao, L., Paramonov, M., Hong, J., Prisle, N. L., Malila, J., Romakkaniemi, S., Lehtinen, K. E. J., Laaksonen, A., Kulmala, M., Massoli, P., Onasch, T. B., Donahue, N. M., Riipinen, I., Davidovits, P., Worsnop, D. R., Petäjä, T., and Virtanen, A.: Adsorptive uptake of water by semisolid secondary organic aerosols, Geophys. Res. Lett., 42, 3063–3068, doi:10.1002/2015GL063142, 2015.

Saukko, E., Lambe, A. T., Massoli, P., Koop, T., Wright, J. P., Croasdale, D. R., Pedernera, D. A., Onasch, T. B., Laaksonen, A., Davidovits, P., Worsnop, D. R., and Virtanen, A.: Humidity-dependent phase state of SOA particles from biogenic and anthropogenic precursors, Atmos. Chem. Phys., 12, 7517–7529, doi:10.5194/acp-12-7517-2012, 2012.

Wagner, R., Möhler, O., Saathoff, H., and Schnaiter, M.: Enhanced high-temperature ice nucleation ability of crystallized aerosol particles after pre-activation at low temperature, J. Geophys. Res. Atmos., 119, 8212-8230, doi:10.1002/2014JD021741, 2014.

---

## Author Comment (AC2) · 11 May 2018

Reply to Referee 2:

We would like to thank the reviewer for her/his comments, which helped to greatly improve the manuscript!
Please find our replies to the comments (which are typeset in bold) below:

**The manuscript by Frey and co-workers investigates the ice nucleation ability of secondary organic aerosol (SOA) particles under so-called mixed-phase cloud conditions. SOA particles have been found to nucleate ice under so-called cold cloud conditions found at high atmospheric altitudes, an ability which is often ascribed to their glassy solid phase state, but their ability for cloud formation at much higher temperatures /lower altitudes has not yet been proven. This work is well-written and addresses this very relevant problem by connecting an aerosol formation vessel (Manchester Aerosol Chamber, MAC) and a vessel for probing the ice nucleation ability of particles (Manchester Ice Cloud Chamber, MICC) in an innovative experimental setup. The authors tested an array of different SOA particles by using three different precursors: one from biogenic and two from anthropogenic origin, making the work relevant for the ambient atmosphere and hence publication in ACP. The main result of this work is that none of the investigated organic aerosols facilitated ice nucleation in the observed temperature and RH region. Such negative results are important to direct future works on the relevance of these particles for cloud formation. However, in my understanding, there was no possibility that ice nucleation should be observed due to the way the experiments were operated. This greatly diminishes the value of this publication and might disincentivize further research on the topic if it won't be stressed clearly by the authors why ice nucleation did not occur. I would like to outline my thoughts in the following and already encourage the author's to disagree with me in their rebuttal in case I misunderstood the experimental setup. At this point, however, I cannot recommend the manuscript for publication.**

The reviewer stresses a very important point and we are confident that we will be able to convince the reviewer to support the publication of our manuscript.

Generally, we tried to follow an experimental path that is as closely aligned with processes in nature as possible: The generation of SOA was performed at conditions which are close to realistic conditions in the lower boundary layer, where precursor gases for SOA formation are emitted (i.e. room temperature and relative humidities wrt water of 45-50%). Previous studies often did not follow such a realistic path, generating SOA at below 10% humidity, some even down to almost 0% humidity (e.g. Prenni et al., 2009, Baustian et al., 2013, Wagner et al., 2017). From ground the aerosol is lifted to regions, where mixed-phase clouds potentially form, implicating cooling of the aerosol and changes in humidity. The transfer of the aerosol in our experiments from the aerosol to the cloud chamber and the subsequent cooling of the transferred air including the aerosol could be seen to mimic those conditions. Then, first a water cloud is formed (therefore, water saturation is needed) from which ice nucleation could take place to form a mixed-phase cloud. This is achieved by quasi-adiabatically cooling of the chamber by performing a chamber expansion.

The modelling study by Shiraiwa et al., 2017 suggests that SOA already is in a semi-solid state over the mid-latitudes and semi-arid regions as Europe, India, Australia, Mexico, and western US. Going up to higher altitudes, the particle phase state becomes more frequently solid or semi-solid. According to Shiraiwa et al. almost all SOA will be solid at 500hPa, corresponding to about 5.5km or roughly -20°C (according to standard atmosphere). And -20°C is the starting temperature of our experiments. They mention convective uplifting of the air in their model, however, whether the RH follows a similar trajectory as in our measurements, we cannot say. Also Mikhailov et al. (2009), based on

measurements of hydration and dehydration with a hygroscopicity tandem differential mobility analyzer (H-TDMA), conclude that the (semi-) solid phases should not only be important at high altitudes but also influence cloud processes at the relative humidities and temperatures of the lower troposphere and the boundary layer. Therefore, our measurements seem relevant as some of the tested systems are known/shown to be semi-solid after formation in the aerosol chamber (see bounce measurements, and also Saukko et al., 2012, and Pajunoja et al, 2015) and we conduct our experiments under these conditions where, according to the above mentioned studies, SOA should exist in (semi-) solid phase state and behave in a way that influences cloud processes, as cloud formation and ice nucleation.

Our measurements start at (close to) water saturation, which facilitates a quick start of cloud formation. Furthermore, ice saturation is needed in order to nucleate ice, thus, being at ice supersaturation is prerequisite as well. According to Koop's (schematic not representational) phase diagram (Koop et al., 2011), all particles (including the amorphous solid) are deliquesced at 90% $RH_w$. So the RH at which we operated in no way diminishes the general applicability of the results.

**General critique**
**In the following I will assume that a glassy phase state is necessary for SOA particles to nucleate ice. If this is the case, a look at the phase diagram of glassy organic material should have sufficed to predict the lacking ice nucleation ability of the SOA particles observed in these experiments (e.g. Koop et al., 2011). The authors state that the MICC was at water saturation before expansion and during the filling of the vessel. At water saturation, a hygroscopic substance cannot be in a viscous or glassy phase state, no matter the glass transition temperature of the pure substance.**

According to Koop's schematical phase diagram (as mentioned above) all particles are deliquesced at about 70% RH (amorphous solid should be deliquesced at about 60%). Given that the schematic is correct and glassy aerosol needed for ice to nucleate from SOA, Ignatius et al., 2016 should not have observed ice nucleation at roughly -39°C and Sice of 1.3 (coldest example of their figure 5, showing alpha-pinene experiments), as RHw would be about 89% in that case. Either, the particles in Ignatius et al.'s study were not glassy, or the schematic phase diagram might have a different look for varied SOA species.

If we assume water saturation in mixed-phase clouds (and according to e.g. Ansmann et al., (2009) the ice phase in mixed-phase clouds is observed after the liquid phase had been established) your argument above would mean that there will never be ice nucleation from SOA in any mixed-phase cloud. This should be tested as hypothesis in a future study. On the other hand, Shiraiwa et al. (2017) and Mikhailov et al. (2009) suggest that these particles should have an impact on cloud processes in the lower troposphere. Thus, we think that our experiments in the way they were conducted are relevant.

**In previous chamber experiments showing the ice nucleation ability of organic aerosols (e.g. Murray et al. (2010), Wilson et al. (2012)), expansion was always initialized at around ice saturation, thus far below water saturation level. The only reason particles could be in a viscous state in the experiment at hand would be if they would cool faster than they could take up water upon entering MICC and hence maintain a non-equilibrium state. These dynamics however seem very difficult to predict since the exact temperature and humidity profiles between MAC and MICC are probably difficult to obtain. Hence, the authors investigated in this study not only if organic aerosols could induce ice nucleation at higher**

**temperatures than expected, but also when initialized at higher humidities than in previous studies. Ice nucleation by organic aerosols under mixed-phase cloud conditions should only be possible if the glassy state is somehow maintained despite the high temperatures and humidities.**
Please see reply above. Correct, most studies generate their SOA particles in very dry conditions, except Wang et al. (2012), who also generated their SOA in more humid conditions at 35±5%.

**In general, the authors miss to mention that not only temperature is important to determine aerosol phase state, but also humidity. A study by Berkemeier et al. (2014) investigating the kinetics of water uptake of viscous organic particles shows that there is a sensitive interplay between temperature, initial humidity and humidification rate that enables organic aerosols to be in the glassy state at humidities relevant for ice nucleation.**
The reviewer has a good point here; we now mention the importance of relative humidity for the aerosol phase state in according places (mainly introduction and discussion).

**If it was not the goal to use the glassy phase state of organic aerosols to trigger ice nucleation this has to be stated more clearly in this manuscript. In this case however, the discussion devoted to bounciness of particles would be superfluous (or even misleading) and a different incentive to discuss organic aerosols as ice nuclei would have to be presented. It is hard to imagine ice nucleation on liquid aerosol particles.**
First of all, we aimed at characterising the ice nucleation (or lack thereof) in terms of SOA phase state. Here, we show that deposition nucleation is not happening to the SOA particles under the conditions of our experiments. Also immersion freezing does not happen to the SOA generated in our experiments, which means that there are no efficient solid/insoluble IN inclusions in the liquid droplets we see. As stated in Berkemeier et al. (2014), solid inclusions in liquid SOA particles can well be realistic in the atmosphere (we added to the discussion accordingly):
"Berkemeier et al. (2014) further state that humidification in typical atmospheric updrafts (or cloud chamber experiments) may be fast enough to cause a difference in phase state from that of equilibrium, as the time for diffusion of the water into the particle is longer than the time for humidification. This can result in a particle that has a liquid outer shell but still contains solid inclusions/a solid core even at RH above the quasi-equilibrium glass transition. Thus, SOA partciles could potentially act as immersion freezing nuclei in conditions where they are supposed to be liquid, if in equilibrium."

**Specific remarks**
**• In the introduction, it seems worth mentioning that many more studies have investigated ice formation from organic aerosols. Examples here are works from the Knopf group (e.g. Wang et al., 2012; Charnawskas et al., 2017), the Tolbert group (Baustian et al., 2013) or at the AIDA chamber (e.g. Wilson et al., 2012). Overall the number of references in this paper is very lacking and too much focused on the author's own work.**
We gave the introduction a major overhaul and also revised the discussion to include more references and compare to the referenced works.

**• As mentioned above, the introduction too much focused on temperatures, too little on relative humidities. The biggest problem with ice nucleation and glassy SOA are the high relative humidities necessary to nucleate ice on the particles, which in turn leads to non-viscous phase states. A discussion on this competition**

**process can be found in Berkemeier et al. (2014)."p. 4, l.11 – Out of curiosity, why would the oxidation be conducted with ozone and not also under irradiation with UV light to generate OH radicals? Most products of SOA formation should not be susceptible to reaction with ozone."**

We have moved the section about 'Particle bounce measurements' from the supplement into the main manuscript. These measurements show that TMB and alpha-pinene do exist in a semi-solid phase state even at high relative humidities. Heptadecane indeed is in a liquid phase even at very low relative humidities. We enhanced the discussion to account for the effect of relative humidity on phase state.

Regarding your question about the cleaning procedure: We regularly also perform SOA background experiments, which include UV radiation. Based on experience with the chamber, we found that regular weekly background experiments are sufficient to keep the chamber walls clean enough, such that the daily cleaning (overnight) between two experiments can be performed without irradiation. This combination of cleaning and background cycles furthermore allows enough time to run the actual SOA experiments and not stretch the time frame for experiments too far. We have expanded the cleaning section in this regard.

**• The usage of the words "bouncy" and "non-bouncy" in this study are misleading. What the author's should say are the conditions under which these particles bounce (RH and T). Do they bounce at -20∘C and water saturation? Hence, in the context of p.8, l.23, this is not a meaningful statement.**

Considering the comments by the other reviewer, we changed wording here since bouncy/non-bouncy is not really a physical parameter. We now use the phase state definitions instead (in our experiments 'semi-solid' and 'liquid'). Furthermore, we moved the section about 'Particle bounce measurements' from the supplement into the main manuscript. From those measurements it becomes clear that heptadecane is in a liquid phase and alpha-pinene and TMB are semi-solid even at high RH, see reply above.

**• The effects of cloud processing in the TMB vs heptadecane SOA experiment are an interesting finding. Is the picture of TMB SOA showing lower activated fractions in the second chamber evacuation run consistent among repetitions of the experiment?**

There was, unfortunately, only one TMB experiment day. Thus, we only have those evacuations shown.

***Minor and technical comments***
***• p. 4, l.2 – boiling point of water***
Done.

***• p. 4, l.11 – Out of curiosity, why would the oxidation be conducted with ozone and not also under irradiation with UV light to generate OH radicals? Most products of SOA formation should not be susceptible to reaction with ozone.***
See our reply to this point above (section starting with "As mentioned above, the introduction ..."

***• p.5, l.11 – A closing parenthesis ")" is missing here.***
Parenthesis added.

***• p.5, l.17 – It is not entirely clear what is meant with "to achieve this". I assume it refers to the cleaning of the chamber, not the particle formation described in the previous sentence.***
It actually refers to the particle formation. We rephrased the sentences:

"To prepare the system for injection of relevant gases for particle formation in MAC the volatile organic compound (VOC) injection glass bulb is heated and flushed with nitrogen. The precursor gases for the SOA and NOx are then injected during the last filling of the MAC air bag."

**• p. 7, l.6 – I believe the word "one" is superfluous here.**
The sentence should use the singular of "particles" as there really only was one particle imaged by the CPI.
"Only one spherical particle was imaged by the CPI."

**• p.9, l.12 – Not the precursors are bouncy, the SOA from these precursors are, at a specific RH and T.**
Rephrased: "The aerosol particles formed from these precursors are both in a semi-solid phase state, i.e. more viscous than the heptadecane particles at the same temperature and relative humidity."

**• p. 9, l.13 – The statement confuses kinetics and thermodynamics, please clarify that not a higher supersaturation is needed to activate viscous particles, it just takes longer time.**
We rephrased: "Thus, it takes a longer time for these viscous precursors to take on water vapour and grow due to diffusion limitations."

*References:*
Ansmann et al., Evolution of the ice phase in tropical altocumulus: SAMUM lidar observations over Cape Verde, J. Geophys. Res., 114, D17208, doi:10.1029/2008JD011659, 2009.

Baustian, K. J., Wise, M. E., Jensen, E. J., Schill, G. P., Freedman, M. A., and Tolbert, M. A.: State transformations and ice nucleation in amorphous (semi-)solid organic aerosol, Atmos. Chem. Phys., 13, 5615–5628, doi:10.5194/acp-13-5615-2013, 2013.

Berkemeier, T., Shiraiwa, M., Pöschl, U., and Koop, T.: Competition between water uptake and ice nucleation by glassy organic aerosol particles, Atmos. Chem. Phys., 14, 12 513–12 531, doi:10.5194/acp-14-12513-2014, 2014.

Koop, T., Bookhold, J., Shiraiwa, M., and Poschl, U.: Glass transition and phase state of organic compounds: dependency on molecular properties and implications for secondary organic aerosols in the atmosphere, Phys. Chem. Chem. Phys., 13, 19 238–19 255, doi:10.1039/C1CP22617G, 2011.

Mikhailov, E., Vlasenko, S., Martin, S. T., Koop, T., and Pöschl, U.: Amorphous and crystalline aerosol particles interacting with water vapor: conceptual framework and experimental evidence for restructuring, phase transitions and kinetic limitations, Atmos. Chem. Phys., 9, 9491–9522, doi:10.5194/acp-9-9491-2009, 2009.

Pajunoja, A., Lambe, A. T., Hakala, J., Rastak, N., Cummings, M. J., Brogan, J. F., Hao, L., Paramonov, M., Hong, J., Prisle, N. L., Malila, J., Romakkaniemi, S., Lehtinen, K. E. J., Laaksonen, A., Kulmala, M., Massoli, P., Onasch, T. B., Donahue, N. M., Riipinen, I., Davidovits, P., Worsnop, D. R., Petäjä, T., and Virtanen, A.: Adsorptive uptake of water by semisolid secondary organic aerosols, Geophys. Res. Lett., 42, 3063–3068, doi:10.1002/2015GL063142, 2015.

Prenni, A. J., Petters, M. D., Faulhaber, A., Carrico, C. M., Ziemann, P. J., Kreidenweis, S. M., and DeMott, P. J.: Heterogeneous ice nucleation measurements of secondary organic aerosol generated from ozonolysis of alkenes, Geophys. Res. Lett., 36, L06808, doi:10.1029/2008GL036957, 2009.

Saukko, E., Lambe, A. T., Massoli, P., Koop, T., Wright, J. P., Croasdale, D. R., Pedernera, D. A., Onasch, T. B., Laaksonen, A., Davidovits, P., Worsnop, D. R., and Virtanen, A.: Humidity-dependent phase state of SOA particles from biogenic and anthropogenic precursors, Atmos. Chem. Phys., 12, 7517–7529, doi:10.5194/acp-12-7517-2012, 2012.

Shiraiwa, M., Li, Y., Tsimpidi, A. P., Karydis, V. A., Berkemeier, T., Pandis, S. N., Lelieveld, J., Koop, T., and Pöschl, U.: Global distribution of particle phase state in atmospheric secondary organic aerosols, Nat. Commun., 8, 15 002, doi:10.1038/ncomms15002, 2017.

Wagner, R., Höhler, K., Huang, W., Kiselev, A., Möhler, O., Mohr, C., Pajunoja, A., Saathoff, H., Schiebel, T., Shen, X., and Virtanen, A.: Heterogeneous ice nucleation of alpha-pinene SOA particles before and after ice cloud processing, J. Geophys. Res. Atmos., 122, 4924–4943, doi:10.1002/2016JD026401, 2016JD026401, 2017.

Wang, B., Lambe, A. T., Massoli, P., Onasch, T. B., Davidovits, P., Worsnop, D. R., and Knopf, D. A.: The deposition ice nucleation and immersion freezing potential of amorphous secondary organic aerosol: Pathways for ice and mixed-phase cloud formation, J. Geophys. Res. Atmos., 117, D16209, doi:10.1029/2012JD018063, d16209, 2012.

---

## Author Comment (AC4) · 11 May 2018

**1 Clean bag transfer**

For a clean bag transfer, both chambers are cleaned according to the respective cleaning procedures. Then a transfer is performed by  refilling MICC from the MAC bag. Aerosol numbers larger than the numbers in the chambers before transfer are thus a result of introduction from leakages in the system. These background aerosol need to be taken into account for any experiment performed on specified aerosol  photochemically produced in the aerosol chamber. That means, in case of ice formation in very low numbers, nucleation of ice on these contaminant aerosol cannot be ruled out.

Aerosol concentrations in the aerosol and cloud chamber were both below $1\,\mathrm{cm}^{-3}$ prior to transfer. After the transfer $10.1\,\mathrm{cm}^{-3}$ aerosol particles were observed by the CPC in the cloud chamber.

[Figure]

Figure S.1: First cloud  activation run after the clean bag transfer (experiment 1, see Table 1 in main manuscript). The panels show the time series of FSSP measurements of size distribution and mean volume diameter (MVD, panel a), total water content (TWC) and number concentration (N, panel b), and temperature and pressure (panel c) during evacuation.

[Figure]

Figure S.2: Second cloud  activation run after the clean bag transfer (experiment 1, see Table 1 in main manuscript). Panels as  in Fig. S.1.

[Figure]

Figure S.3: Third cloud  activation run after the clean bag transfer (experiment 1, see Table 1 in main manuscript). Panels as  in Fig. S.1.

**2**

~~Previous to our measurements, transfer experiments on some of the same (and some additional) systems were performed and the phase state of particles were determined by particle bounce measurements (see Saukko et al., 2012a,b for a detailed desription of the methods). Here an upgraded version of the bounce system, the Aerosol Bounce Instrument (ABI), was employed. ABI consists of a particle size selection unit (neutralizer containing bipolar [210]Po strip and Vienna type long DMA), a humidification unit (Permapure PD 240 12SS, Nafion multitube), impactor unit (MOUDI stage #14 with upstream pressure, $p_{initial} = 0.85$ bar, and downstream pressure, $p_{final} = 0.7$ bar, leading to a cut off aerodynamic diameter $d_a = 67.09$ nm), and two CPCs (TSI, model 3010) for measuring the particle number concentration before and after the impactor. ABI determines a bounced fraction (BF) of particles which is used as an indicator of the phase state of particles; particles with BF $\tilde{0}$ are mechanically liquid whereas particles with $0.1 <$ BF $< 1$ are mechanically solid or semi-solid. Calculation and calibration of bounce measurements are described in more detail by Saukko et al. (2012b). Composition and bounced fraction of SOA particles in MAC. Open symbols represent data collected in experiments where there was no quantitative ice nucleation data. Panel a shows the mass fraction of the AMS fragments measured at m/z=44 throughout their growth in the aerosol chamber. Panel b shows bounced fractions for heptadecane, $\alpha$-pinene/limonene and limonene experiments as a function of RH at the latest available time before transfer. In general higher bounce fractions are found at lower relative humidity. It can clearly be seen that SOA from the heptadecane experiment exhibit very different behaviour, maintaining a bounced fraction of less than 1.2% even at the lowest available RH. All other bounced fraction data is similar to the other examples shown. The heptadecane experiment also shows a much lower m/z=44 fraction than in other experiments.~~

**2 SOA background**

Two types of SOA backgrounds were transferred,  with varying levels of UV radiation. That means, all  chemical substances as in a normal SOA experiment were used, without the actual precursor. In the  first background experiment filters were installed in front of the lamps responsible for photochemical reactions to only allow tropospheric UV radiation to illuminate the chamber, whereas in the  second background experiment, the  filters were removed, allowing hard UV light into the chamber. This second background was used before SOA experiments with heptadecane and TMB precursors, as aerosol is harder to form from these precursors and strong UV light fosters the SOA generation. Also ozone is added in the second SOA background experiment, thus, it can also be seen as a harsh cleaning experiment at the same time.

[Figure]

Figure S.4: First cloud  activation run during  first SOA background measurements (experiment 3, see Table 1 in main manuscript). The uppermost panel shows the SMPS size distributions obtained before the cloud expansion (panel a), followed by time series of FSSP measurements of size distribution and mean volume diameter (MVD, panel b), total water content (TWC) and number concentration (N, panel c), and temperature and pressure (panel d) during evacuation.

[Figure]

Figure S.5: As above for the second cloud  activation run in the  first SOA background experiment (experiment 3, see Table 1 in main manuscript).

[Figure]

Figure S.6: First cloud  activation run on 'harsh UV' SOA background (experiment 5, see Table 1 in main manuscript), panels as  in Fig. S.4.

[Figure]

Figure S.7: Second cloud  activation run on 'harsh UV' SOA background (experiment 5, see Table 1 in main manuscript), panels as  in Fig. S.4.

**3    SOA experiments**

In the following all cloud activation runs performed during the measurement period are shown, except those already shown in the main manuscript.

[Figure]

Figure S.8:  First cloud  activation run performed on SOA from $\alpha$-pinene  precursor (experiment 4, see Table 1 in main manuscript), panels as  in Fig. S.4.

[Figure]

Figure S.9: Second cloud  activation run performed on SOA from $\alpha$-pinene  precursor (experiment 4, see Table 1 in main manuscript), panels as  in Fig. S.4.

[Figure]

Figure S.10: First cloud  activation run performed on SOA from heptadecane precursor (experiment 6, see Table 1 in main manuscript), panels as  in Fig. S.4.

[Figure]

Figure S.11: Second cloud  activation run performed on SOA from heptadecane pre-cursor (experiment 6, see Table 1 in main manuscript), panels as  in Fig. S.4.

[Figure]

Figure S.12: First cloud  activation run performed on SOA from TMB precursor (experiment 7, see Table 1 in main manuscript), panels as in Fig. S.4.

[Figure]

Figure S.13: Second cloud  activation run performed on SOA from TMB precursor (experiment 7, see Table 1 in main manuscript), panels as  in Fig. S.4.

[Figure]

Figure S.14: First cloud  activation run performed on $\alpha$-pinene aerosol precursor (experiment 8, see Table 1 in main manuscript), panels as in Fig. S.4. The second cloud  activation run from this experiment is shown in main manuscript.

[Figure]

Figure S.15: First cloud  activation run performed on SOA from heptadecane precursor (experiment 10, see Table 1 in main manuscript), panels as  in Fig. S.4.  The second cloud  activation run from this experiment is shown in main manuscript.

**4 Control experiments with ammonium sulfate**

[Figure]

Figure S.16: First cloud  activation run performed on ammonium sulfate aerosol (experiment 2, see Table 1 in main text), panels as  in Fig. S.4.

[Figure]

Figure S.17: Second cloud  activation run performed on ammonium sulfate aerosol (experiment 2, see Table 1 in main text), panels as in Fig. S.4.

[Figure]

Figure S.18: Third cloud activation run performed on ammonium sulfate aerosol (experiment 2, see Table 1 in main text), panels as in Fig. S.4.

[Figure]

Figure S.19: First cloud  activation run performed on ammonium sulfate aerosol (experiment 9, see Table 1 in main text), panels as in Fig. S.4.

[Figure]

Figure S.20: Second cloud  activation run performed on ammonium sulfate aerosol (experiment 9, see Table 1 in main text), panels as  in Fig. S.4.

**5 Sensitivity experiment with kaolinite**

First cloud evacuation performed on dust, panels as above.

[Figure]

Figure S.21: Second cloud evacuation activation run performed on dust (experiment 11, see Table 1 in main manuscript), panels as above in Fig. S.4.
The first cloud activation run from this experiment is shown in main manuscript.

Table 1: Mean mode diameters (MMD) of the aerosol size distributions before the cloud activation runs.

|       | system            | run # | MMD [nm]    |
|-------|-------------------|-------|-------------|
| Exp 2 | ammonium sulfate  | 1     | data faulty |
|       |                   | 2     | 49.7        |
|       |                   | 3     | 50.4        |
| Exp 3 | SOA background    | 1     | 37.5        |
|       |                   | 2     | 38.8        |
| Exp 4 | $\alpha$-pinene   | 1     | 244.4       |
|       |                   | 2     | 250.4       |
| Exp 5 | SOA background    | 1     | 92.5        |
|       |                   | 2     | 92.5        |
| Exp 6 | heptadecane       | 1     | 455.9       |
|       |                   | 2     | 455.9       |
| Exp 7 | TMB               | 1     | 187.8       |
|       |                   | 2     | 201.7       |
| Exp 8 | $\alpha$-pinene   | 1     | 204.2       |
|       |                   | 2     | 209.2       |
| Exp 9 | ammonium sulfate  | 1     | 33.7        |
|       |                   | 2     | 39.5        |
| Exp 10| heptadecane       | 1     | 376.3       |
|       |                   | 2     | 371.8       |
| Exp 11| dust (kaolinite)  | 1     | 16.7        |
|       |                   | 2     | 14.1        |

**6**

~~Saukko, E., Lambe, A. T., Massoli, P., Koop, T., Wright, J. P., Croasdale, D. R., Pedernera, D. A., Onasch, T. B., Laaksonen, A., Davidovits, P., Worsnop, D. R., and Virtanen, A.: Humidity-dependent phase state of SOA particles from biogenic and anthropogenic precursors. Atmos. Chem. Phys., 12, 7517–7529, doi:10.5194/acp-12-7517-2012, 2012a.Saukko, E., Kuuluvainen, H., and Virtanen, A.: A Method to resolve the phase state of aerosol particles, Atmos. Meas. Tech., 5, 259–265, doi:10.5194/amt-5-259-2012, 2012b.~~